# DilateQuant: Accurate and Efficient Diffusion Quantization via Weight Dilation

## Abstract

Diffusion models have shown excellent performance on various image generation tasks, but the substantial computational costs and huge memory footprint hinder their low-latency applications in real-world scenarios. Quantization is a promising way to compress and accelerate models. Nevertheless, due to the wide range and time-varying activations in diffusion models, existing methods cannot maintain both accuracy and efficiency simultaneously for low-bit quantization. To tackle this issue, we propose DilateQuant, a novel quantization framework for diffusion models that offers comparable accuracy and high efficiency. Specifically, we keenly aware of numerous unsaturated in-channel weights, which can be cleverly exploited to reduce the range of activations without additional computation cost. Based on this insight, we propose Weight Dilation (WD) that maximally dilates the unsaturated in-channel weights to a constrained range through a mathematically equivalent scaling. WD costlessly absorbs the activation quantization errors into weight quantization. The range of activations decreases, which makes activations quantization easy. The range of weights remains constant, which makes model easy to converge in training stage. Considering the time-varying activations, we design a Temporal Parallel Quantizer (TPQ), which sets time-step quantization parameters and supports parallel quantization for different time steps by utilizing an indexing approach, significantly improving the performance and reducing time cost. To further enhance performance while preserving efficiency, we introduce a Block-wise Knowledge Distillation (BKD) to align the quantized models with the full-precision models at a block level. The simultaneous training of time-step quantization parameters and weights minimizes the time required, and the shorter backpropagation paths decreases the memory footprint of the quantization process. Extensive experiments demonstrate that DilateQuant significantly outperforms existing methods in terms of accuracy and efficiency.

## 1 Introduction

Recently, diffusion models have shown excellent performance on image generation (Li et al., 2022; Zhang et al., 2023b;c), but the substantial computational costs and huge memory footprint hinder their low-latency applications in real-world scenarios. Numerous methods (Nichol & Dhariwal, 2021; Song et al., 2020; Lu et al., 2022) have been proposed to find shorter sampling trajectories for the thousand iterations of the denoising process, effectively reducing latency. However, complex networks with a large number of parameters used in each denoising step are computational and memory intensive, which slow down inference and consume high memory footprint. For instance, the Stable-Diffusion (Rombach et al., 2022) with 16GB of running memory still takes over one second to perform one denoising step, even on the high-performance A6000.

Model quantization is one of the most popular compression methods. By quantizing the weights and activations with low-bit integers, we can reduce memory requirements and accelerate computational operations. The effects become more noticeable as the bit-width decreases. For example, employing 8-bit models can achieve up to a $4\times$ memory compression and $2.35\times$ speedup compared to 32-bit full-precision models on a T4 GPU (Kim et al., 2022). Adopting 4-bit models can further deliver an additional $2\times$ compression and $1.59\times$ speedup compared to 8-bit models. Thus, quantization is a highly promising way to facilitate the low-latency applications of diffusion models on source-constrained hardware.

Typically, existing quantization techniques are implemented through two main approaches: Post-Training Quantization (PTQ) and Quantization-Aware Training (QAT). As shown in Figure 1, PTQ (Liu et al., 2024) calibrates the quantization parameter with a small calibration dataset and does not rely on end-to-end retraining, making it data- and time-efficient. However, it brings severe performance degradation at low bit-width. In contrast, QAT (Esser et al., 2019) can maintain performance at lower bit-width, but it requires retraining the whole model, which is time-consuming and resource-intensive. For instance, when applying both standard approaches to DDIM (Song et al., 2020) on CIFAR-10, QAT (Esser et al., 2019) results in a $3.3\times$ increase in GPU memory footprint (9.97 GB vs. 3.01 GB) and an $14.3\times$ extension of quantization time (13.89 GPU-hours vs. 0.97 GPU-hours) compared to PTQ (Liu et al., 2024). Due to the huge gap in time cost and GPU consumption, PTQ is more preferred despite the fact that QAT outperforms PTQ.

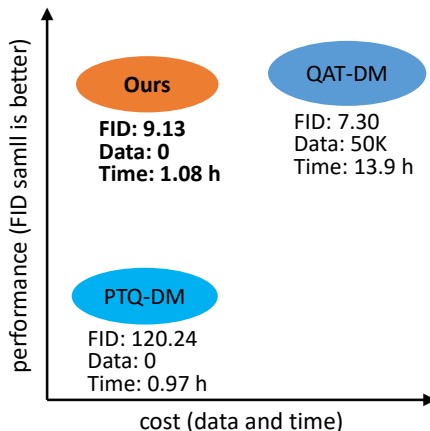

Figure 1: An overview of the cost-vs-performance trade-off across various approaches. Data is collected from DDIM with 4-bit quantization on CIFAR-10.

Unfortunately, while previous methods (Xiao et al., 2023c; Li & Gu, 2023; Xiao et al., 2023b; Li et al., 2023b) of quantization have achieved remarkable success in single-time networks, the wide range and time-varying activations caused by the unique temporal network of diffusion models make them fail. Specifically, since the diffusion models infer in pixel space or latent space, the absence of layer normalization results in a wide range of activations, complicating activation quantization. For example, in the same UNet network, the range of activations is almost $2.5\times$ larger than that of the segmentation models (Ronneberger et al., 2015), as shown in Figure 2(a). Equivalent scaling techniques address the wide range of activations by shifting the quantization difficulty from activations to weights. Some methods (Xiao et al., 2023a; Shao et al., 2023; Lin et al., 2024; Zhang et al., 2023a) utilizing equivalent scaling have shown success in large language models (LLMs) by tackling outliers in certain channels, but these methods are not appropriate for diffusion models, where outliers exist in all channels, as shown in Figure 2(b). Unconstrained scaling of all outlier channels significantly alters the weight range, making it difficult for model to converge in training stage. In addition, the temporal network induces a highly dynamic distribution of activations that varies across time steps, as shown in Figure 2(c), further diminishing the performance of quantization. Numerous PTQ methods (Li et al., 2023a; Liu et al., 2024) have been explored to enhance results based on the properties of diffusion models, none of them break through the 6-bit quantization for activations. And the QAT methods(Esser et al., 2019) retrain the whole model separately for each time step using the original datasets, which is not practical due to the significant time and resources.

In this paper, we propose DilateQuant, a novel quantization framework that can achieve QAT-like performance with PTQ-like efficiency. Specifically, we propose a weight-aware equivalent scaling algorithm, called Weight Dilation (WD), which searches for unsaturated in-channel weights and dilates them to the boundary of the quantized range, using the max-min values of the out-channel weights as constraints. WD narrows the range of activations while keeping the weights range unchanged, making activation quantization easier and ensuring model convergence during the training stage. This approach effectively alleviates the wide range activations. To address the difficulty of quantization for time-varying activations, previous methods (He et al., 2023; Wang et al., 2024) set multiple activation quantizers for one layer and trains them individually using different time-step calibration sets, which is data- and time-inefficient. On the other hand, we design a Temporal Parallel Quantizer (TPQ), which sets time-step quantization parameters and supports parallel quantization for different time steps by utilizing an indexing approach, significantly improving performance and training efficiency, as evidenced by a $160\times$ reduction in calibration and a $2\times$ reduction in training time compared to the SoTA method (He et al., 2023) for DDIM on CIFAR-10. To further enhance performance while preserving efficiency, we introduce a Block-wise Knowledge Distillation (BKD) to avoids data- and time-consuming retraining of the whole model, distilling the full-precision model to its quantized counterpart at block level using a data-free approach. Additionally, it further minimizes the time and memory footprint required by training the time-step quantization parameters simultaneously and using the shorter backpropagation paths, respectively.

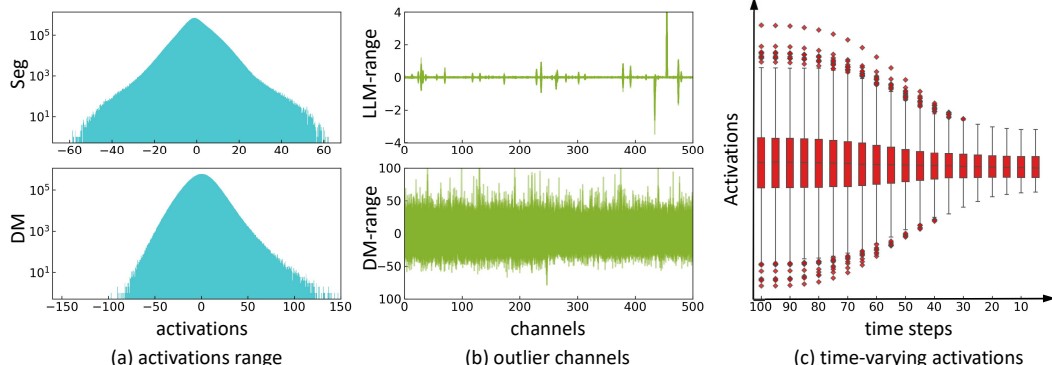

(a) activations range    (b) outlier channels    (c) time-varying activations

Figure 2: (a) showcases a wider range of activations in diffusion model (DM) compared to segmentation model (Seg). (b) demonstrates the different outlier challenges for DM and LLM. (c) shows the dynamic distribution activations of DM. The activations of DM and Seg are from the first block output of the upsample stage of UNet network. The activations of LLM come from the output of the penultimate layer.

The contributions of our works are summarized as follows. 1) We formulate a novel quantization framework for diffusion models, DilateQuant, which offers comparable accuracy and high efficiency. 2) The WD and TPQ address the wide range and time-varying activations for diffusion models. And the BKD efficiently enhances performance. 3) Through extensive experiments, we demonstrate that DilateQuant outperforms existing methods across lower quantization settings (6-bit, 4-bit), various models (DDPM, LDM-4, LDM-8, Stable-Diffusion), and different datasets (CIFAR-10, LSUN-Bedroom, LSUN-Church, ImageNet, MS-COCO). The reproduction of Dilate-Quant is robust and easy as no hyper-parameters are introduced.

## 2 RELATED WORK

### 2.1 DIFFUSION MODEL ACCELERATION

While diffusion models have generated high-quality images, the substantial computational costs and huge memory footprint hinder their low-latency applications in real-world scenarios. To reduce the inference computation, numerous methods have been proposed to find shorter sampling trajectories, efficiently accelerating the denoising process. For example, (Nichol & Dhariwal, 2021) shortens the denoising steps by adjusting variance schedule; (Song et al., 2020) generalizes diffusion process to a non-Markovian process by modifying denoising equations; (Lu et al., 2022) uses high-order solvers to approximate diffusion generation. These methods have achieved significant success, obtaining comparable performance with nearly 10% of the denoising steps. However, they involve expensive retraining and complex computations. Conversely, we focus on the complex networks of diffusion models, accelerating them at each denoising step with a quantization method, which not only reduces the computational cost but also compresses the model size.

### 2.2 MODEL QUANTIZATION

Model quantization, which represents the original floating-point parameters with low-bit values, compresses model size and accelerates inference. Depending on whether the model's weights are fine-tuned or not, it generally falls into two categories: Post-Training Quantization (PTQ) and Quantization-Aware Training (QAT). PTQ calibrates the quantization parameters with a small dataset and does not require fine-tuning the model's weights, making it data- and time-efficient. The reconstruction-based PTQ techniques, such as BRECQ (Li et al., 2021), utilize gradient descent algorithms to optimize quantization parameters, which have yielded remarkable results in conventional models. Nevertheless, the unique temporal networks of diffusion models cause them to fail. To address the issues, PTQ4DM (Shang et al., 2023) and Q-diffusion (Li et al., 2023a) design a specialized calibration dataset, and EDA-DM (Liu et al., 2024) refines the reconstruction loss. Although these PTQ methods enhance results based on the properties of diffusion models, none of

them break through the 6-bit quantization. On the other hand, QAT retrains the whole model after the quantization operation, maintaining performance at lower bit-width. However, the significant training resources (original dataset, training time, and GPU consumption) make it not practical for diffusion models. For instance, the recent work TDQ (So et al., 2023) requires 200K training iterations on a 50K original dataset. To efficiently quantize diffusion models to lower precision, EfficientDM (He et al., 2023) fine-tunes all of the model's weights with an additional LoRA module, while QuEST (Wang et al., 2024) selectively trains some sensitive layers. Unfortunately, although they achieve 4-bit quantization of the diffusion models, both of them are non-standard (please refer to Appendix E for detail). Hence, the standard quantization of low-bit diffusion models with high accuracy and efficiency is still an open question.

## 3 PRELIMINARIES

### 3.1 QUANTIZATION

The uniform quantizer is one of the most hardware-friendly choices, and we use it in our work. The quantization-dequantization process of it can be defined as:

$$Quant : \boldsymbol{x}_{int} = clip\left(\left\lfloor\frac{\boldsymbol{x}}{\Delta}\right\rceil + z, 0, 2^b - 1\right) \tag{1}$$

$$DeQuant : \hat{\boldsymbol{x}} = \Delta \cdot (\boldsymbol{x}_{int} - z) \approx \boldsymbol{x} \tag{2}$$

where $\boldsymbol{x}$ and $\boldsymbol{x}_{int}$ are the floating-point and quantized values, respectively, $\lfloor\cdot\rceil$ represents the rounding function, and the bit-width $b$ determines the range of clipping function $clip(\cdot)$. In the dequantization process, the dequantized value $\hat{\boldsymbol{x}}$ approximately recovers x. Notably, the upper and lower bounds of $\boldsymbol{x}$ determine the quantization parameters: scale factor $\Delta$ and zero-point $z$, as follows:

$$\Delta = \frac{max(\boldsymbol{x}) - min(\boldsymbol{x})}{2^b - 1}, \quad z = \left\lfloor\frac{-min(\boldsymbol{x})}{\Delta}\right\rceil \tag{3}$$

Combining the two processes, we can provide a general definition for the quantization function, $Q(\boldsymbol{x})$, as:

$$Q(\boldsymbol{x}) = \Delta \cdot \left(clip\left(\left\lfloor\frac{\boldsymbol{x}}{\Delta}\right\rceil + z, 0, 2^b - 1\right) - z\right) \tag{4}$$

As can be seen, quantization is the process of introducing errors: $\lfloor\cdot\rceil$ and $clip(\cdot)$ result in rounding error ($E_{round}$) and clipping error ($E_{clip}$), respectively. To set the quantization parameters, we commonly use two calibration methods: Max-Min and MSE. For the former, quantization parameters are calibrated by the max-min values of $\boldsymbol{x}$, eliminating the $E_{clip}$, but resulting in the largest $\Delta$; for the latter, quantization parameters are calibrated with appropriate values, but introduce the $E_{clip}$.

### 3.2 EQUIVALENT SCALING

Equivalent scaling is a mathematically equivalent per-channel scaling transformation that offline shifts the quantization difficulty from activations to weights. For a linear layer in diffusion model, the output $\boldsymbol{Y} = \boldsymbol{XW}, \boldsymbol{Y} \in \mathbb{R}^{N \times C^o}, \boldsymbol{X} \in \mathbb{R}^{N \times C^i}, \boldsymbol{W} \in \mathbb{R}^{C^i \times C^o}$, where $N$ is the batch-size, $C^i$ is the input channel, and $C^o$ is the output channel. The activation $\boldsymbol{X}$ divides a per-in-channel scaling factor $\boldsymbol{s} \in \mathbb{R}^{C^i}$, and weight $\boldsymbol{W}$ scales accordingly in the reverse direction to maintain mathematical equivalence:

$$\boldsymbol{Y} = (\boldsymbol{X}/\boldsymbol{s})(\boldsymbol{s} \cdot \boldsymbol{W}) \tag{5}$$

The formula also suits the conv layer. By ensuring that $\boldsymbol{s} > \boldsymbol{1}$, the range of activations can be made smaller and the range of weights larger, thus in transforming the difficulty of quantization from activations to weights. In addition, given that the $\boldsymbol{X}$ is usually produced from previous linear operations, we can easily fuse the scaling factor into previous layers' parameters offline so as not to introduce additional computational overhead in the inference. Currently, equivalent scaling is primarily used in the quantization of LLMs to smooth out activation outliers in certain channels. While some methods (Xiao et al., 2023a; Lin et al., 2024; Shao et al., 2023; Zhang et al., 2023a) have achieved success in LLMs, they fail in diffusion models due to different quantization challenges, please see Appendix G for details.

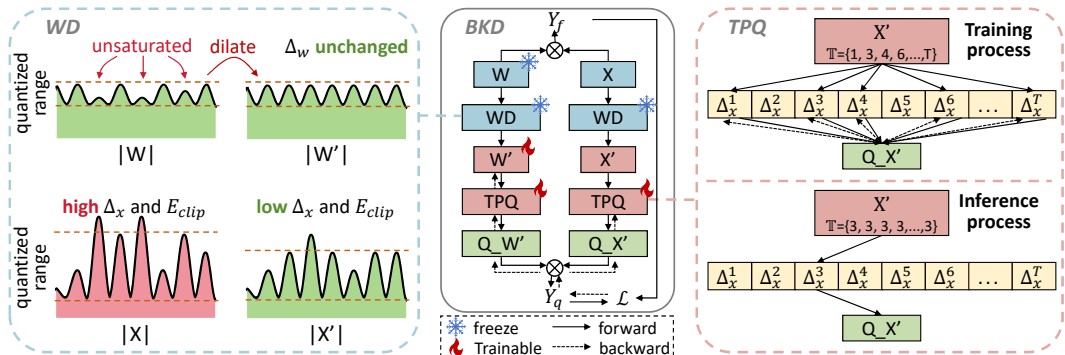

Figure 3: An overview of DilateQuant. WD narrows the activations range while maintaining the weights range unchanged. TPQ sets time-step quantization parameters and supports parallel training. BKD aligns the quantized network with the full-precision network at block level.

## 4 METHOD

### 4.1 WEIGHT DILATION

**Analyzing quantization error.** We start by analyzing the error from weight-activation quantization. Taking a linear layer with $X \in \mathbb{R}^{N \times C^i}$ and $W \in \mathbb{R}^{C^i \times C^o}$ as example, considering that we calibrate the quantization parameters of $X$ and $W$ with a MSE and Max-Min manner, respectively, the quantization function (Eq. 4) for activations and weights can be briefly written as:

$$Q(X) = \Delta_x \cdot clip\left(\left\lfloor \frac{X}{\Delta_x} \right\rceil\right), \quad Q(W) = \Delta_w \cdot \left\lfloor \frac{W}{\Delta_w} \right\rceil \tag{6}$$

where $\Delta_x$ and $\Delta_w$ are scale factors for activations and weights, respectively. Thus, the quantization error can be defined as:

$$E(X, W) = \|XW - Q(X)Q(W)\|_F \tag{7}$$

where $\|\cdot\|_F$ denotes Frobenius Norm. The formula can be further decomposed as:

$$E(X, W) \leq \|X\|_F \|W - Q(W)\|_F + \|X - Q(X)\|_F (\|W\|_F + \|W - Q(W)\|_F) \tag{8}$$

Please see Appendix 6 for the proof. Ultimately, the quantization error is influenced by four elements–the magnitude of the weight and activation, $\|W\|_F$ and $\|X\|_F$, and their respective quantization errors, $\|W - Q(W)\|_F$ and $\|X - Q(X)\|_F$. Furthermore, the $\|W - Q(W)\|_F$ and $\|X - Q(X)\|_F$ result from rounding (denoted as $E_{round}$) and cliping (denoted as $E_{clip}$) function, and they can be represented in finer granularity as:

$$\|X - Q(X)\|_F = \Delta_x \cdot (E_{round} + E_{clip}), \quad \|W - Q(W)\|_F = \Delta_w \cdot E_{round} \tag{9}$$

Since the rounding function maps a floating-point number to an integer, $E_{round}$ does not vary, as demonstrated in AWQ (Lin et al., 2024). Previous methods scale the $X$ and $W$ using a simply scaling factor $s \in \mathbb{R}^{C^i}$, which consider both the magnitudes of activations and weights, to obtain the scaled $X'$ and $W'$. The quantization functions and errors after scaling are as follows:

$$Q(X') = Q(X/s) = \Delta_x' \cdot clip\left(\left\lfloor \frac{X/s}{\Delta_x'} \right\rceil\right), \qquad Q(W') = Q(s \cdot W) = \Delta_w' \cdot \left\lfloor \frac{s \cdot W}{\Delta_w'} \right\rceil \tag{10}$$

$$\|X - Q(X)\|_F' = \Delta_x' \cdot (E_{round} + E_{clip}'), \qquad \|W - Q(W)\|_F' = \Delta_w' \cdot E_{round} \tag{11}$$

where $\Delta_x'$ and $\Delta_w'$ are new scale factors, and $E_{clip}'$ is the new error of cliping function. By ensuring that $s > 1$, which results in $E_{clip}'/E_{clip} < 1$, $\Delta_x'/\Delta_x < 1$, and $\Delta_w'/\Delta_w > 1$ (according to Eq. 3), the $\|X - Q(X)\|_F$ and $\|X - Q(X)\|_F$ decrease while the $\|W\|_F$ and $\|W - Q(W)\|_F$ equivalently increasing. Consequently, there are no overall change in $E(X, W)$ and the excessive disruption of the initial weight range hinders the model's ability to converge during training. Therefore, the perfect scaling we desired is to decrease activations range while maintaining weights range.

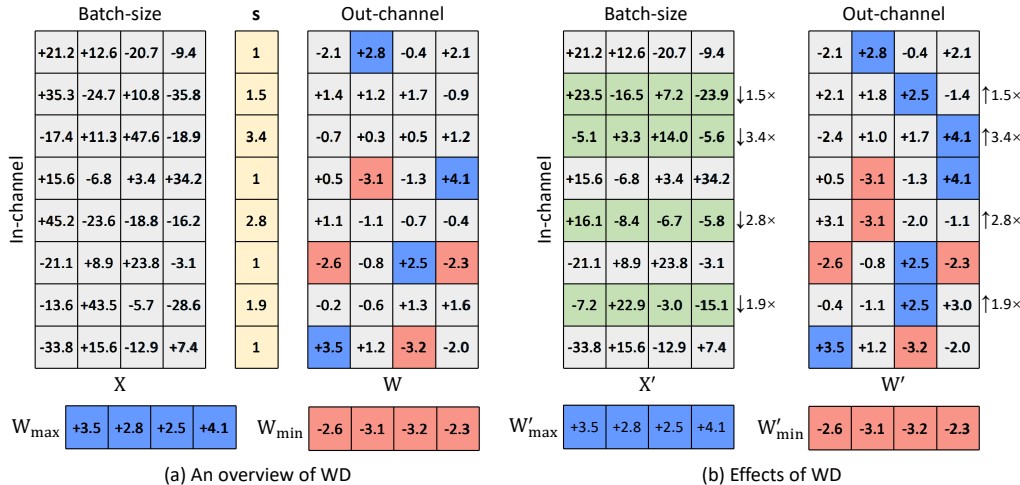

Figure 4: (a) WD searches for unsaturated in-channel weights and determines scaling factor completely dependent on the max-min values of each out-channel of the weights. (b) WD alleviates the wide range activations by dilating unsaturated channels to a constrained range.

**Searching channel for scaling.** Given that the dimension of weights quantization is per-out-channel and the dimension of scaling is per-in-channel, we ensure the max-min values ($W_{max} \in \mathbb{R}^{C^o}$, $W_{min} \in \mathbb{R}^{C^o}$) of each out-channel unchanged and record their indexes of in-channel to form a set $A$. For example, the $A$ in Figure 4(a) is $\{1,4,6,8\}$. Iterating through the index of in-channel $k \in \{1, \ldots, C^i\}$, if $k \in A$, we set $s_k = 1$, representing no scaling; if $k \notin A$, the $W_k$ denotes as unsaturated in-channel weights, and we set $s_k$ by dilating $W_k$ to $W_{max}$ or $W_{min}$:

$$s_{k1} = min(W_{max}/W_k.clamp(min = \epsilon)) \tag{12}$$

$$s_{k2} = min(W_{min}/W_k.clamp(max = -\epsilon)) \tag{13}$$

$$s_k = min(s_{k1}, s_{k2}) \tag{14}$$

where $\epsilon = 1e - 5$ and $clamp$ function specify the range of the $k_{th}$ in-channel of weight $W_k$, $s_{k1}$ and $s_{k2}$ denote the maximum $s$ with $W_{max}$ and $W_{min}$ as constraints, respectively. Consequently, as shown in Figure 4(b), we maximize $s > 1$ while keeping $W'_{max} = W_{max}$ and $W'_{min} = W_{min}$. The workflow and effects of WD are detailed in Appendix F.

## 4.2 TEMPORAL PARALLEL QUANTIZER

Previous methods (He et al., 2023; Wang et al., 2024) utilize multiple activation quantizers for a layer to quantize activations at different time steps. However, since each quantizer is independent, these methods optimize each quantizer individually using time-step calibration sets, which is data- and time-inefficient. For example, EfficientDM uses 819.2K samples for a total of 12.8K iterations for DDIM on CIFAR-10 (Krizhevsky et al., 2009).

Different from previous methods, as shown in Figure 3, we design a novel quantizer, denotes as Temporal Parallel Quantizer (TPQ), which sets time-step quantization parameters for activations, instead of simply stacking quantizers. Specifically, it utilizes *an indexing approach* to call the corresponding quantization parameters for samples at different time steps. This enables support for parallel training of different quantization parameters, significantly reducing the data and time costs of training. For a model with $T$ time steps, the quantization parameters of TPQ are as follows:

$$\Delta_x = \{\Delta_x^1, \Delta_x^2, \Delta_x^3, \ldots, \Delta_x^T\}, \quad z_x = \{z_x^1, z_x^2, z_x^3, \ldots, z_x^T\} \tag{15}$$

We detail TPQ design for the different layers of the diffusion models. For the conv and linear layers, they take input $x \in \mathbb{R}^{|\mathbb{T}| \times C^i}$ and $x \in \mathbb{R}^{|\mathbb{T}| \times C^i \times H \times W}$, respectively, where $\mathbb{T}$ is a set containing different time-step indexes, $\mathbb{T} \subset \{1, \ldots, T\}$, $|\cdot|$ represents the number of set elements. The quantization operation of them can be represented as:

$$Q(x) = \Delta_x^{\mathbb{T}} \cdot \left( clip \left( \left\lfloor \frac{x}{\Delta_x^{\mathbb{T}}} \right\rceil + z_x^{\mathbb{T}}, 0, 2^b - 1 \right) - z_x^{\mathbb{T}} \right) \tag{16}$$

where $\Delta_x^{\mathbb{T}}$ and $z_x^{\mathbb{T}}$ denote the quantization parameters corresponding to $\mathbb{T}$, respectively. For the attention layers, they take input $x \in \mathbb{R}^{|\mathbb{T}*H| \times C^i \times M}$ due to the $H$ heads, where "$*$" represents cat operation and $M$ is the number of tokens. So the quantization parameters in Eq. 16 are replaced with $\Delta_x^{\mathbb{T}*H}$ and $z_x^{\mathbb{T}*H}$, respectively.

## 4.3 BLOCK-WISE KNOWLEDGE DISTILLATION

QAT significantly alleviates accuracy degradation in low-bit cases, but it has several limitations for diffusion models: (1) QAT typically requires original training data, which can sometimes be challenging or even impossible to obtain due to privacy or copyright concerns; (2) QAT involves end-to-end retraining of the whole complex networks, which is training-unstable and time-intensive.

To address these limitations, inspired by the reconstruction method in PTQ (Li et al., 2021), we propose a novel distillation strategy called Block-wise Knowledge Distillation (BKD). Assume the target model for quantization has K blocks $(B_1, \ldots, B_K)$, and the input samples of model are $x$, which is generated by the full-precision model. BKD trains the quantized network block-by-block and aligns it with full-precision network at block level. More specifically, assume that block $B_k$ is going to be quantized, and its quantized version is $\hat{B}_k$. We update the quantization parameters $(\Delta_x^{\mathbb{T}}, z_x^{\mathbb{T}}, \Delta_w)$ and weights $(w)$ of $\hat{B}_k$ using the mean square loss $\mathcal{L}$:

$$\mathcal{L}_{\Delta_x^{\mathbb{T}}, z_x^{\mathbb{T}}, \Delta_w, w} = MSE\left(B_k \cdot B_{k-1} \cdot B_{k-2} \cdot \ldots \cdot B_1(x) - \hat{B}_k \cdot \hat{B}_{k-1} \cdot \hat{B}_{k-2} \cdot \ldots \cdot \hat{B}_1(x)\right) \quad (17)$$

As can be seen, (1) BKD does not rely on original training data; (2) BKD shortens the gradient back-propagation path by aligning blocks, which enhances training stability and decreases the memory footprint of the quantization process. In addition, BKD trains quantization parameters and weights in parallel, which not only further saves training time but adapts the weights to each time step.

Table 1: Results of unconditional image generation. The "Calib." presents the number of calibration samples and "Prec. (W/A)" indicates the bit-width. $\star$ denotes our implementation according to open-source codes and $\dagger$ represents results directly obtained by rerunning open-source codes.

| Task | Method | Calib. | Prec. (W/A) | TBops | Size (MB) | FID ↓ | sFID ↓ | IS ↑ |
|---|---|---|---|---|---|---|---|---|
| | FP | - | 32/32 | 6.2 | 143.0 | 4.26 | 4.46 | 9.03 |
| CIFAR-10 | EDA-DM $\star$ | 5120 | 6/6 | 0.2 | 27.0 | 26.68 | 14.10 | **9.35** |
| $32 \times 32$ | EfficientDM $\dagger$ | 1.6384M | 6/6 | 0.2 | 27.0 | 17.29 | 9.38 | 8.85 |
| | DilateQuant | 5120 | 6/6 | 0.2 | 27.0 | **4.46** | **4.64** | 8.92 |
| DDPM | EDA-DM $\star$ | 5120 | 4/4 | 0.1 | 18.1 | 120.24 | 36.72 | 4.42 |
| steps = 100 | EfficientDM $\dagger$ | 1.6384M | 4/4 | 0.1 | 18.1 | 81.27 | 30.95 | 6.68 |
| | DilateQuant | 5120 | 4/4 | 0.1 | 18.1 | **9.13** | **6.92** | **8.56** |
| | FP | - | 32/32 | 98.4 | 1317.4 | 3.02 | 7.21 | 2.29 |
| LSUN-Bedroom | EDA-DM $\star$ | 5120 | 6/6 | 3.5 | 247.8 | 10.56 | 16.22 | 2.12 |
| (Yu et al., 2015) | EfficientDM $\dagger$ | 102.4K | 6/6 | 3.5 | 247.8 | 5.43 | 15.11 | 2.15 |
| $256 \times 256$ | QuEST $\dagger$ | 5120 | 6/6 | 3.5 | 247.8 | 10.1 | 19.57 | **2.20** |
| | DilateQuant | 5120 | 6/6 | 3.5 | 247.8 | **3.92** | **8.90** | 2.17 |
| LDM-4 | EDA-DM $\star$ | 5120 | 4/4 | 1.6 | 165.5 | N/A | N/A | N/A |
| steps = 100 | EfficientDM $\dagger$ | 102.4K | 4/4 | 1.6 | 165.5 | 15.27 | 19.87 | 2.11 |
| eta = 1.0 | QuEST $\dagger$ | 5120 | 4/4 | 1.6 | 165.5 | N/A | N/A | N/A |
| | DilateQuant | 5120 | 4/4 | 1.6 | 165.5 | **8.99** | **14.88** | **2.13** |
| | FP | - | 32/32 | 19.1 | 1514.5 | 4.06 | 10.89 | 2.70 |
| LSUN-Church | EDA-DM $\star$ | 5120 | 6/6 | 0.7 | 284.9 | 10.76 | 18.23 | 2.43 |
| (Yu et al., 2015) | EfficientDM $\dagger$ | 102.4K | 6/6 | 0.7 | 284.9 | 6.92 | 12.84 | 2.65 |
| $256 \times 256$ | QuEST $\dagger$ | 5120 | 6/6 | 0.7 | 284.9 | 6.83 | 11.93 | 2.65 |
| | DilateQuant | 5120 | 6/6 | 0.7 | 284.9 | **5.33** | **11.61** | 2.66 |
| LDM-8 | EDA-DM $\star$ | 5120 | 4/4 | 0.3 | 190.3 | N/A | N/A | N/A |
| steps = 100 | EfficientDM $\dagger$ | 102.4K | 4/4 | 0.3 | 190.3 | 15.08 | 16.53 | **2.67** |
| eta = 0.0 | QuEST $\dagger$ | 5120 | 4/4 | 0.3 | 190.3 | 13.03 | 19.50 | 2.63 |
| | DilateQuant | 5120 | 4/4 | 0.3 | 190.3 | **10.10** | **16.22** | 2.62 |

Table 2: Quantization results of conditional image generation.

| Task | Method | Calib. | Prec. (W/A) | TBops | Size (MB) | FID ↓ | sFID ↓ | CLIP ↑ |
|------|--------|--------|-------------|-------|-----------|-------|--------|--------|
| MS-COCO (Lin et al., 2014) $512 \times 512$ Stable-Diffusion steps = 50 eta = 0.0 scale = 7.5 | FP | - | 32/32 | 347.2 | 4112.5 | 21.96 | 33.86 | 26.88 |
| | EDA-DM [*] | 512 | 6/6 | 12.4 | 772.8 | N/A | N/A | N/A |
| | EfficientDM [*] | 12.8K | 6/6 | 12.4 | 772.8 | 154.61 | 74.50 | 19.01 |
| | DilateQuant | 512 | 6/6 | 12.4 | 772.8 | **24.69** | **33.06** | **26.62** |
| | EDA-DM [*] | 512 | 4/4 | 5.6 | 515.9 | N/A | N/A | N/A |
| | EfficientDM [*] | 12.8K | 4/4 | 5.6 | 515.9 | 216.43 | 111.76 | 14.35 |
| | DilateQuant | 512 | 4/4 | 5.6 | 515.9 | **44.82** | **42.97** | **23.51** |

| Task | Method | Calib. | Prec. (W/A) | TBops | Size (MB) | FID ↓ | sFID ↓ | IS ↑ |
|------|--------|--------|-------------|-------|-----------|-------|--------|------|
| ImageNet (Deng et al., 2009) $256 \times 256$ LDM-4 steps = 20 eta = 0.0 scale = 3.0 | FP | - | 32/32 | 102.3 | 1824.6 | 11.69 | 7.67 | 364.72 |
| | EDA-DM [*] | 1024 | 6/6 | 3.7 | 343.2 | 11.52 | 8.02 | **360.77** |
| | EfficientDM [†] | 102.4K | 6/6 | 3.7 | 343.2 | 8.69 | 8.10 | 309.52 |
| | QuEST [†] | 5120 | 6/6 | 3.7 | 343.2 | 8.45 | 9.36 | 310.12 |
| | DilateQuant | 1024 | 6/6 | 3.7 | 343.2 | **8.25** | **7.66** | 312.30 |
| | EDA-DM [*] | 1024 | 4/4 | 1.7 | 229.2 | 20.02 | 36.66 | 204.93 |
| | EfficientDM [†] | 102.4K | 4/4 | 1.7 | 229.2 | 12.08 | 14.75 | 122.12 |
| | QuEST [†] | 5120 | 4/4 | 1.7 | 229.2 | 38.43 | 29.27 | 69.58 |
| | DilateQuant | 1024 | 4/4 | 1.7 | 229.2 | **8.01** | **13.92** | **257.24** |

## 5 EXPERIMENT

### 5.1 EXPERIMENTAL SETUP

**Models and metrics.** The comprehensive experiments include DDPM, LDM (Song et al., 2020; Rombach et al., 2022) and Stable-Diffusion on 5 datasets. The performance of the quantized models is evaluated with FID (Heusel et al., 2017), sFID (Salimans et al., 2016), IS (Salimans et al., 2016), and CLIP score (Hessel et al., 2021). Following the common practice, the Stable-Diffusion generates 10,000 images, while all other models generate 50,000 images. Besides, we also calculate the Bit Operations and Size of models to visualize the effects of model acceleration and compression.

**Quantization and comparison settings.** We employ DilateQuant with the standard channel-wise quantization for weights and layer-wise quantization for activations. To highlight the efficiency, DilateQuant selects 5120 samples for calibration and trains for 5K iterations with a batch size of 32, aligning with PTQ-based method (Liu et al., 2024). The Adam (Kingma & Ba, 2014) optimizer is adopted, and the learning rates for quantization parameters and weights are set as 1e-4 and 1e-2, respectively. For the experimental comparison, we compare DilateQuant with PTQ-based method (Liu et al., 2024) and variant QAT-based methods (He et al., 2023; Wang et al., 2024). Since these two variant QAT-based methods employ non-standard settings, we modify them to standard settings for a fair comparison. To further compare with them, we also employ the same non-standard settings on DilateQuant to conduct experiments in the Appendix E. All experiments are performed on one RTX A6000. The more detailed experimental implementations are showcased in Appendix B.

### 5.2 MAIN RESULT

**Unconditional generation.** We focus on the performance of low-bit quantization to highlight the advantages of DilateQuant. As reported in Table 1, in 4-bit quantization, previous works all suffer from non-trivial performance degradation. For instance, EDA-DM and QuEST become infeasible on LSUN-Bedroom, and EfficientDM remains far from practical usability on LSUN-Church. In sharp contrast, DilateQuant achieves a substantial improvement in quantization performance, with encouraging 6.28 and 4.98 FID improvement over EfficientDM on two LSUN datasets, respectively. Additionally, in 6-bit quantization, DilateQuant can achieve a fidelity comparable to that of the full-precision baseline.

**Conditional generation.** The quantization results for conditional generation are reported in Table 2. For text-guided generation with 6-bit precision, DilateQuant improves the FID to 24.69 with 5.3× Model size compression and 27.9× Bit Operations reduction, effectively advancing the low-

latency applications of Stable-Diffusion in real-world scenarios. Besides, DilateQuant achieves significant improvements at all bit-width settings on class-guided generation. We add human preference assessments in Appendix I.

Table 3: The efficacy of different component proposed in this paper.

| Method | | | | Prec. | Time Cost | GPU Memory | | | |
|---|---|---|---|---|---|---|---|---|---|
| WD | TPQ | BKD | Framework | (W/A) | (hours) | (MB) | FID↓ | sFID↓ | IS↑ |
| ✗ | ✗ | ✗ | PTQ | 4/4 | 0.97 | 3019 | 120.24 | 36.72 | 4.42 |
| ✗ | ✓ | ✗ | PTQ | 4/4 | 0.97 | 3278 | 31.49 | 17.95 | 7.67 |
| ✓ | ✗ | ✗ | PTQ | 4/4 | 1.08 | 3076 | 26.26 | 16.73 | 7.78 |
| ✓ | ✓ | ✗ | PTQ | 4/4 | 1.08 | 3439 | 16.27 | 11.83 | 8.09 |
| ✗ | ✗ | ✓ | QAT | 4/4 | 0.98 | 3019 | 18.45 | 11.53 | 8.67 |
| ✗ | ✓ | ✓ | QAT | 4/4 | 0.98 | 3278 | 9.63 | 7.08 | 8.45 |
| ✓ | ✗ | ✓ | QAT | 4/4 | 1.08 | 3076 | 9.66 | 7.06 | 8.58 |
| ✓ | ✓ | ✓ | QAT | 4/4 | 1.08 | 3439 | 9.13 | 6.92 | 8.56 |

## 5.3 ABLATION STUDY

The ablation experiments are conducted over DDIM on CIFAR-10 with 4-bit quantization. We start by analysing the efficacy of each proposed component, as reported in Table 3. We use the SoTA PTQ-based framework, EDA-DM (Liu et al., 2024), as the baseline, which fails to maintain performance. By incorporating WD and TPQ, we push the performance limits of PTQ methods to achieve an FID score of 16.27. The introduction of BKD transforms the approach into a QAT framework, as it involves retraining the quantized weight of models. By combining BKD, DilateQuant reduces the FID score to 9.13, achieving a generation quality comparable to that of full-precision models.

Table 4: Efficiency comparisons of various quantization frameworks.

| Task | Method | Framework | Calib. | Training Data | Time Cost | GPU Memory | FID↓ |
|---|---|---|---|---|---|---|---|
| CIFAR-10 32 × 32 | EDA-DM | PTQ | 5120 | 0 | 0.97 h | 3019 MB | 120.24 |
| | LSQ | QAT | - | 50K | 13.89 h | 9974 MB | 7.30 |
| | EfficientDM | V-QAT | 1.6384M | 0 | 2.98 h | 9546 MB | 81.27 |
| | Ours | V-QAT | 5120 | 0 | 1.08 h | 3439 MB | 9.13 |
| ImageNet 256 × 256 | QuEST | V-QAT | 5120 | 0 | 15.25 h | 20642 MB | 38.43 |
| | Ours | V-QAT | 1024 | 0 | 6.56 h | 14680 MB | 8.01 |

We also conduct the efficiency analysis of DilateQuant by comparing it with PTQ (Liu et al., 2024), QAT (Esser et al., 2019), and variant QAT (He et al., 2023; Wang et al., 2024) methods. As reported in Table 4, the PTQ method fails to maintain performance and the QAT method requires significant resources. In sharp contrast, DilateQuant achieves QAT-like accuracy with PTQ-like time cost and GPU memory. The efficiency comparisons on other models are reported in Appendix D. We also add the ablation experiments of DilateQuant for time steps and samplers in Appendix C.

## 6 CONCLUSION

In this work, we propose DilateQuant, a novel quantization framework for diffusion models that offers comparable accuracy and high efficiency. Specifically, we find the unsaturation property of the in-channel weights and exploit it to alleviate the wide range of activations. By dilating the unsaturated channels to a constrained range, our method costlessly absorbs the activation quantization errors into weight quantization. Furthermore, we design a flexible quantizer that sets time-step quantization parameters to time-varying activations and supports parallel quantization for training process, significantly improving the performance and reducing time cost. We also introduce a novel knowledge distillation strategy to enhance performance, which aligns the quantized models with the full-precision models at a block level. The simultaneous training of parameters and shorter backpropagation paths minimize the time and memory footprint required. Exhaustive experiments demonstrate that DilateQuant significantly outperforms existing methods in low-bit quantization.

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

DILATEQUANT: SUPPLEMENTARY MATERIALS

P.1

$$
\begin{aligned}
E(X, W) =& \|XW - Q(X)Q(W)\|_F \\
=& \|XW - XQ(W) + XQ(W) - Q(X)Q(W)\|_F \\
\leq& \|X(W - Q(W))\|_F + \|(X - Q(X))Q(W)\|_F \\
\leq& \|X\|_F \|W - Q(W)\|_F + \|X - Q(X)\|_F \|Q(W)\|_F \\
\leq& \|X\|_F \|W - Q(W)\|_F + \|X - Q(X)\|_F \|W - (W - Q(W))\|_F \\
\leq& \|X\|_F \|W - Q(W)\|_F + \|X - Q(X)\|_F \left(\|W\|_F + \|W - Q(W)\|_F\right)
\end{aligned}
\tag{18}
$$

## A  SUPPLEMENTARY MATERIAL INTRODUCTION

In this supplementary material, we present the correlative introductions and some experiments mentioned in the paper. The following items are provided:

- Detailed experimental implementations for all experiments in Appendix B.
- Robustness of DilateQuant for time steps and samplers in Appendix C.
- Efficiency comparisons of various quantization frameworks in Appendix D
- Thorough comparison with EfficientDM and QuEST in Appendix E.
- Workflow and effects of Weight Dilation algorithm in Appendix F.
- Different equivalent scaling algorithms for diffusion models in Appendix G.
- Hardware-Friendly quantization in Appendix H.
- Human preference evaluation in Appendix I.

## B  DETAILED EXPERIMENTAL IMPLEMENTATIONS

In this section, we present detailed experimental implementations, including the pre-training models, qunatization settings, and evaluation.

The DDPM[1] models and LDM[2] models we used for the experiments are obtained from the official websites. For text-guided generation with Stable-Diffusion, we use the CompVis codebase[3] and its v1.4 checkpoint. The LDMs consist of a diffusion model and a decoder model. Following the previous works (Liu et al., 2024; He et al., 2023; Wang et al., 2024), DilateQuant focus only on the diffusion models and does not quantize the decoder models. We empoly channel-wise asymmetric quantization for weights and layer-wise asymmetric quantization for activations. The input and output layers of models use a fixed 8-bit quantization, as it is a common practice. The weight and activation quantization ranges are initially determined by minimizing values error, and then optimized by our knowledge distillation strategy to align quantized models with full-precision models at block level. Since the two compared methods employ non-standard settings, we modify them to standard settings for a fair comparison. More specifically, we quantize all layers for EfficientDM, including `Upsample`, `Skip_connection`, and `AttentionBlock`'s qkvw, which lack quantization in open-source code[4]. However, when these layers, which are important for quantization, are added, the performance of EfficientDM degrades drastically. To recover performance, we double the number of training iterations. QuEST utilizes channel-wise quantization for activations at 4-bit precision in the code[5], which is not supported by hardware. Therefore, we adjust the quantization setting to layer-wise quantization for activations. For experimental evaluation, we use open-source tool *pytorch-OpCounter*[6] to calculate the Size and Bops of models before and after quantization.

---

[1] https://github.com/ermongroup/ddim
[2] https://github.com/CompVis/latent-diffusion
[3] https://github.com/CompVis/stable-diffusion
[4] https://github.com/ThisisBillhe/EfficientDM
[5] https://github.com/hatchetProject/QuEST
[6] https://github.com/Lyken17/pytorch-OpCounter

And following the quantization settings, we only calculate the diffusion model part, not the decoder and encoder parts. We use the ADM's TensorFlow evaluation suite *guided-diffusion*[7] to evaluate FID, sFID, and IS, and use the open-source code *clip-score*[8] to evaluate CLIP scores. As the per practice (Liu et al., 2024; Wang et al., 2024), we employ the zero-shot approach to evaluate Stable-Diffusion on COCO-val for the text-guided experiments, resizing the generated $512 \times 512$ images and validation images in $300 \times 300$ with the center cropping to evaluate FID score and using text prompts from COCO-val to evaluate CLIP score.

## C   ROBUSTNESS OF DILATEQUANT FOR TIME STEPS AND SAMPLERS

To assess the robustness of DilateQuant for samplers, we conduct experiments over LDM-4 on ImageNet with three distant samplers, including DDIMsampler Song et al. (2020), PLMSsampler Liu et al. (2022), and DPMSolversampler Lu et al. (2022). Given that time step is the most important hyperparameter for diffusion models, we also evaluate DilateQuant for models with different time steps, including 20 steps and 100 steps. As shown in Table 5, our method showcases excellent robustness across different samplers and time steps, leading to significant performance enhancements compared to previous methods. Specifically, our method outperforms the full-precision models in terms of FID and sFID at 6-bit quantization, and the advantages of our method are more pronounced compared to existing methods at the lower 4-bit quantization.

Table 5: The robustness of DilateQuant for time steps and samplers.

| Task | Method | Calib. | Prec. (W/A) | FID ↓ | sFID ↓ | IS ↑ |
|---|---|---|---|---|---|---|
| | FP | - | 32/32 | 11.69 | 7.67 | 364.72 |
| LDM-4 — DDIM time steps = 20 | EDA-DM [⋆] | 1024 | 6/6 | 11.52 | 8.02 | **360.77** |
| | EfficientDM [†] | 102.4K | 6/6 | 8.69 | 8.10 | 309.52 |
| | DilateQuant | 1024 | 6/6 | **8.25** | **7.66** | 312.30 |
| | EDA-DM [⋆] | 1024 | 4/4 | 20.02 | 36.66 | 204.93 |
| | EfficientDM [†] | 102.4K | 4/4 | 12.08 | 14.75 | 122.12 |
| | DilateQuant | 1024 | 4/4 | **8.01** | **13.92** | **257.24** |
| | FP | - | 32/32 | 11.71 | 7.08 | 379.19 |
| LDM-4 — PLMS time steps = 20 | EDA-DM [⋆] | 1024 | 6/6 | 11.27 | 6.59 | **363.00** |
| | EfficientDM [†] | 102.4K | 6/6 | 9.85 | 9.36 | 325.13 |
| | DilateQuant | 1024 | 6/6 | **7.68** | **5.69** | 315.85 |
| | EDA-DM [⋆] | 1024 | 4/4 | 17.56 | 32.63 | 203.15 |
| | EfficientDM [†] | 102.4K | 4/4 | 14.78 | 9.89 | 103.34 |
| | DilateQuant | 1024 | 4/4 | **9.56** | **8.12** | **243.72** |
| | FP | - | 32/32 | 11.44 | 6.85 | 373.12 |
| LDM-4 — DPM-Solver time steps = 20 | EDA-DM [⋆] | 1024 | 6/6 | 11.14 | 7.95 | **357.16** |
| | EfficientDM [†] | 102.4K | 6/6 | 8.54 | 9.30 | 336.11 |
| | DilateQuant | 1024 | 6/6 | **7.32** | **6.68** | 330.32 |
| | EDA-DM [⋆] | 1024 | 4/4 | 30.86 | 39.40 | 138.01 |
| | EfficientDM [†] | 102.4K | 4/4 | 14.36 | 13.82 | 109.52 |
| | DilateQuant | 1024 | 4/4 | **8.98** | **9.97** | **247.62** |
| | FP | - | 32/32 | 4.45 | 6.27 | 238.39 |
| LDM-4 — DDIM time steps = 100 | EDA-DM [⋆] | 1024 | 6/6 | 12.21 | 12.13 | 71.50 |
| | EfficientDM [†] | 102.4K | 6/6 | **5.57** | 7.50 | **165.15** |
| | DilateQuant | 1024 | 6/6 | 5.97 | **7.44** | 162.93 |
| | EDA-DM [⋆] | 1024 | 4/4 | N/A | N/A | N/A |
| | EfficientDM [†] | 102.4K | 4/4 | 20.70 | 11.79 | 72.67 |
| | DilateQuant | 1024 | 4/4 | **9.85** | **10.79** | **147.63** |

---

[7] https://github.com/openai/guided-diffusion
[8] https://github.com/Taited/clip-score

# D  EFFICIENCY COMPARISONS OF VARIOUS QUANTIZATION FRAMEWORKS

We investigate the efficiency of DilateQuant across data resource, time cost, and GPU memory. We compare our method with PTQ-based method (Liu et al., 2024) and variant QAT-based method (He et al., 2023) on the mainstream diffusion models (DDPM, LDM, Stable-Diffusion). As reported in Table 6, our method performs PTQ-like efficiency, while significantly improving the performance of the quantized models. This provides an affordable and efficient quantization process for diffusion models.

Table 6: Efficiency comparisons of various quantization frameworks with 4-bit quantization across data resource, time cost, and GPU memory.

| Model | Method | Calib. | Time Cost (hours) | GPU Memory (MB) | FID ↓ |
|---|---|---|---|---|---|
| DDPM CIFAR-10 | PTQ | 5120 | 0.97 | 3019 | 120.24 |
| | V-QAT | 1.6384M | 2.98 | 9546 | 81.27 |
| | Ours | 5120 | 1.08 | 3439 | 9.13 |
| LDM ImageNet | PTQ | 1024 | 6.43 | 13831 | 20.02 |
| | V-QAT | 102.4K | 5.20 | 22746 | 12.08 |
| | Ours | 1024 | 6.56 | 14680 | 8.01 |
| Stable-Diffusion MS-COCO | PTQ | 512 | 7.23 | 30265 | 236.31 |
| | V-QAT | 12.8K | 30.25 | 46082 | 216.43 |
| | Ours | 512 | 7.41 | 31942 | 42.97 |

# E  THOROUGH COMPARISON WITH EFFICIENTDM AND QUEST

EfficientDM (He et al., 2023) and QuEST (Wang et al., 2024) are two variance QAT-based methods, which achieve 4-bit quantization of the diffusion models with efficiency. However, both of them are non-standard. Specifically, EfficientDM preserves some layers at full-precision, notably the `Upsample`, `Skip_connection`, and the matrix multiplication of `AttentionBlock`'s qkvw. These layers have been demonstrated to have the most significant impact on the quantization of diffusion models in previous works (Shang et al., 2023; Li et al., 2023a; Liu et al., 2024). QuEST employs standard channel-wise quantization for weights and layer-wise quantization for activations at 6-bit precision. However, at 4-bit precision, it uses channel-wise quantization for the activations of all `Conv` and `Linear` layers, which is hardly supported by the hardware because it cannot factor the different scales out of the accumulator summation (please see Appendix H for details), leading to inefficient acceleration.

Table 7: Comparison with EfficientDM and QuEST in both standard and non-standard settings.

| Task | Mode | Method | Prec. (W/A) | Size (MB) | FID ↓ |
|---|---|---|---|---|---|
| LSUN-Church (Yu et al., 2015) $256 \times 256$ LDM-8 steps = 100 eta = 0.0 | - | FP | 32/32 | 1514.5 | 4.06 |
| | **Non-standard** Not quantize for all layers | EfficientDM | 6/6 | 315.0 | 6.29 |
| | | DilateQuant | 6/6 | 315.0 | **4.73** |
| | | EfficientDM | 4/4 | 222.7 | 14.34 |
| | | DilateQuant | 4/4 | 222.7 | **8.68** |
| | **Standard** Quantize for all layers | EfficientDM | 6/6 | 284.9 | 6.92 |
| | | DilateQuant | 6/6 | 284.9 | **5.33** |
| | | EfficientDM | 4/4 | 190.3 | 15.08 |
| | | DilateQuant | 4/4 | 190.3 | **10.10** |
| | **Non-standard** Channel-wise for A | QuEST | 4/4 | 190.3 | 11.76 |
| | | DilateQuant | 4/4 | 190.3 | **8.94** |
| | **Standard** Layer-wise for A | QuEST | 4/4 | 190.3 | 13.03 |
| | | DilateQuant | 4/4 | 190.3 | **10.10** |

To thoroughly compare DilateQuant with EfficientDM and QuEST, we conduct experiments on LSUN-church with standard and non-standard quantization settings. When neglecting these layers that are important for quantization, DilateQuant extremely reduces the FID to 8.68 with 4-bit quantization. Compared to the standard setting, the performance improvement is more noticeable. When setting channel-wise quantization for activations, DilateQuant also reduces a 2.84 FID compared with QuEST. Conclusively, DilateQuant significantly outperforms EfficientDM and QuEST at different quantization precisions for both standard and non-standard settings, which demonstrates the stability and standards of DilateQuant.

## F    WORKFLOW AND EFFECTS OF WEIGHT DILATION ALGORITHM

The comprehensive workflow of Weight Dilation is illustrated in Algorithm 1. We implement WD in three steps: searching unsaturated channels for scaling (Lines 2-3), calculating scaling factor (Lines 5-10), and scaling activations and weights (Line 12). WD alleviates the wide range activations for diffusion models through a novel equivalent scaling algorithm. In addition, all operations of WD can be implemented simply, making it efficient.

---

**Algorithm 1** Overall workflow of WD

---

**Input**: full-precision $\boldsymbol{X} \in \mathbb{R}^{N \times C^i}$ and $\boldsymbol{W} \in \mathbb{R}^{C^i \times C^o}$
**Output**: scaled $\boldsymbol{X}'$ and $\boldsymbol{W}'$.

1: **searching unsaturated channels for scaling:**
2:     obtain $W_{max} \in \mathbb{R}^{C^o}$ and $W_{min} \in \mathbb{R}^{C^o}$
3:     record in-channel indexes of $W_{max}$ and $W_{min}$ as set $A$
4: **calculating scaling factor:**
5:     **for** $k = 1$ to $C^i$ **do**
6:         **if** $k \in A$:
7:             set $\boldsymbol{s}_k = 1$
8:         **else**:
9:             calculate scaling factor $\boldsymbol{s}_k$ with $W_{max}$ and $W_{min}$ as constraints
10:     **end for**
11: **scaling** $X$ and $W$**:**
12:     calculate $\boldsymbol{X}' = X / \boldsymbol{s}$ and $\boldsymbol{W}' = W \cdot \boldsymbol{s}$
13: **return** $\boldsymbol{X}'$ and $\boldsymbol{W}'$

---

We assess the effects of WD on various quantization tasks. As reported in Table 8, WD stably achieves $\boldsymbol{s} > \boldsymbol{1}$ while maintaining $\Delta'_w \approx \Delta_w$. It effectively improves performance at different quantized models by losslessly reducing the activation quantization error.

Table 8: Effects of WD on different tasks with 4-bit quantization.

| Tasks | CIFAR-10 | LSUN-Bedroom | LSUN-Church | ImageNet | MSCOCO |
|---|---|---|---|---|---|
| $\Delta'_w/\Delta_w$ | 1.02 | 1.02 | 1.01 | 1.01 | 1.02 |
| $E'_{clip}/E_{clip}$ | 0.83 | 0.92 | 0.92 | 0.93 | 0.92 |
| proportion of $s > 1$ | 39.2% | 52.4% | 32.8% | 36.5% | 43.8% |
| $\Delta'_x/\Delta_x$ | 0.91 | 0.92 | 0.91 | 0.92 | 0.90 |
| FID $\downarrow$ | 9.13 (-0.50) | 8.99 (-0.25) | 10.10 (-0.20) | 8.01 (-0.27) | 44.82 (-0.79) |

## G    DIFFERENT EQUIVALENT SCALING ALGORITHMS FOR DIFFUSION MODELS

In this section, we start by analyzing the differences between LLMs and diffusion models in terms of the challenges of activation quantization. As shown in Figure 2(b), the activation outliers of the diffusion models are present in all channels, unlike in LLMs where the activation outliers only exist in fixed channels. Additionally, the range of activations for diffusion models is also larger than that of the LLMs. Therefore, it is essential to scale the number of channels as much as possible for the

diffusion models. Some equivalent scaling algorithms are proposed to smooth out the activation outliers in LLMs, and these methods have achieved success. SmoothQuant (Xiao et al., 2023a) scales all channels using a hand-designed scaling factor. AWQ (Lin et al., 2024) only scales a few of channels based on the salient weight. OmniQuant (Shao et al., 2023) proposes a learnable equivalent transformation to optimize the scaling factors in a differentiable manner. DGQ (Zhang et al., 2023a) devises a percentile scaling scheme to select the scaled channels and calculate the scaling factors. OS+ conducts channel-wise shifting and scaling across all channels.

Unfortunately, when we applied methods similar to these previous equivalent scaling algorithms to diffusion models, we find that none of them work. Specifically, we employ these five methods for diffusion models as follows: (1) For the method similar to SmoothQuant, we scale all channels before quantization using a smoothing factor $\alpha = 0.5$; (2) For the method similar to AWQ, we scale 1% of channels based on the salient weight, setting smoothing factor the same as SmoothQuant; (3) For the method similar to OmniQuant, we modify the scaling factors to be learnable variants and train them block by block with a learning rate of 1e-5; (4) For the method similar to DGQ, we scale the top 0.5% of quantization-sensitive channels, setting scaling factor based on the clipping threshold. (5) For OS+, we perform shifting and scaling across all channels, consistent with the original work. However, as shown in Table 9, all of these methods result in higher FID and sFID scores compared to no scaling. The reason for this result is that although the range of activations decreases, the range of weights also increases significantly, making it more difficult for the model to converge during the training stage. In contrast, the Weight Dilation algorithm we proposed scales the number of channels as much as possible. It searches for unsaturated in-channel weights and dilates them to a constrained range based on the max-min values of the out-channel weights. The algorithm reduces the range of activations while maintaining the weights range unchanged. This effectively makes activation quantization easier and ensures model convergence, reducing the FID and sFID scores to 9.13 and 6.92 in 4-bit quantization, respectively.

Table 9: The results of various equivalent scaling algorithms for DDIM on CIFAR-10.

| Prec. | Metrics | No scaling | SmoothQuant | OmniQuant | AWQ | DGQ | OS+ | Ours |
|-------|---------|-----------|-------------|-----------|------|------|------|------|
| W4A4 | proportion of $s > 1$ | 0% | 100% | 100% | 1% | 0.5% | 100% | 39.2% |
| | FID ↓ | 9.63 | 9.99 | 9.86 | 10.34 | 9.72 | 9.78 | **9.13** |
| | sFID ↓ | 7.08 | 7.29 | 7.34 | 7.53 | 7.78 | 7.23 | **6.92** |
| | IS ↑ | 8.45 | 8.46 | 8.50 | 8.38 | 8.52 | 8.36 | **8.56** |
| W6A6 | proportion of $s > 1$ | 0% | 100% | 100% | 1% | 0.5% | 100% | 39.2% |
| | FID ↓ | 5.75 | 5.44 | 5.56 | 5.85 | 5.09 | 5.81 | **4.46** |
| | sFID ↓ | 4.96 | 4.87 | 4.89 | 5.19 | 4.84 | 4.99 | **4.64** |
| | IS ↑ | 8.80 | 8.86 | 8.81 | 8.78 | 8.89 | 8.76 | **8.92** |

# H  HARDWARE-FRIENDLY QUANTIZATION

In this section, we investigate the correlation between quantization settings and hardware acceleration. We start with the principle of quantization to achieve hardware acceleration. A matrix-vector multiplication, $y = Wx + b$, is calculated by a neural network accelerator, which comprises two fundamental components: the processing elements $C_{n,m}$ and the accumulators $A_n$. The calculation operation of accelerator is as follows: firstly, the bias values $b_n$ are loaded into accumulators; secondly, the weight values $W_{n,m}$ and the input values $x_m$ are loaded into $C_{n,m}$ and computed in a single cycle; finally, their results are added in the accumulators. The overall operation is also referred to as Multiply-Accumulate (MAC):

$$A_n = \sum_m W_{n,m} x_m + b_n \tag{19}$$

where $n$ and $m$ represent the out-channel and in-channel of the weights, respectively. The pre-trained models are commonly trained using FP32 weights and activations. In addition to MAC calculations, data needs to be transferred from memory to the processing units. Both of them severely impact the speed of inference. Quantization transforms floating-point parameters into fixed-point parameters, which not only reduces the amount of data transfer but also the size and energy consumption

of the MAC operation. This is because the cost of digital arithmetic typically scales linearly to quadratically with the number of bits, and fixed-point addition is more efficient than its floating-point counterpart. Quantization approximates a floating-point tensor $\boldsymbol{x}$ as:

$$\hat{\boldsymbol{x}} = \Delta \cdot \boldsymbol{x}_{int} \approx \boldsymbol{x} \tag{20}$$

where $\boldsymbol{x}_{int}$ and $\hat{\boldsymbol{x}}$ are integer tensors and quantized tensors, respectively, and $\Delta$ is scale factor.

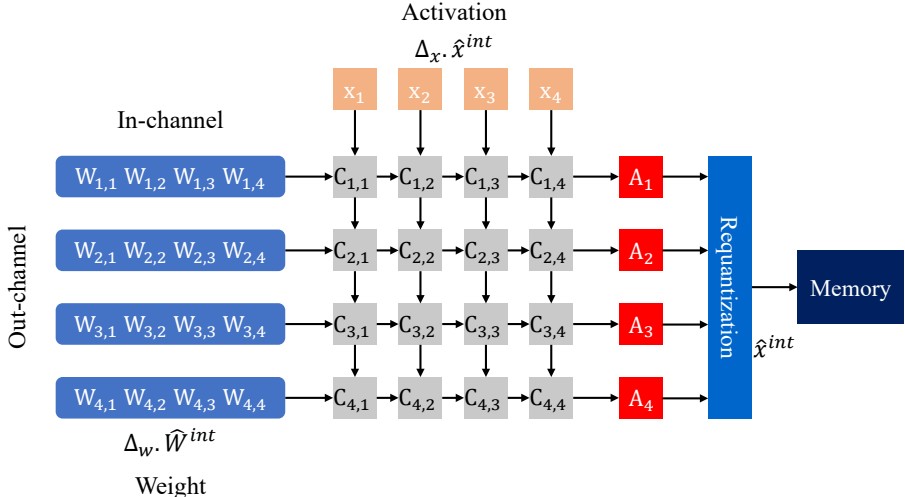

Figure 5: A schematic of matrix-multiply logic in accelerator for quantized inference.

Quantization settings have different granularity levels. Figure 5 shows the accelerator operation after the introduction of quantization. If we set both activations and weights to be layer-wise quantization, the new MAC operation can be represented as:

$$\hat{A}_n = \sum_m \hat{W}_{n,m}\hat{x}_m + b_n$$

$$= \sum_m (\Delta_w \hat{W}_{n,m}^{int})(\Delta_x \hat{x}_m^{int}) + b_n$$

$$= \Delta_w \Delta_x \sum_m \hat{W}_{n,m}^{int} \hat{x}_m^{int} + b_n \tag{21}$$

where $\Delta_w$ and $\Delta_x$ are scale factors for weights and activations, respectively, $\hat{W}_{n,m}^{int}$ and $\hat{x}_m^{int}$ are integer values. The bias is typically stored in higher bit-width (32-bits), so we ignore bias quantization for now. As can be seen, this scheme factors out the scale factors from the summation and performs MAC operations in fixed-point format, which accelerates the calculation process. The activations are quantized back to integer values $\hat{x}_n^{int}$ through a requantization step, which reduces data transfer and simplifies the operations of the next layer.

To approximate the operations of quantization to full-precision, channel-wise quantization for weights is widely used, which sets quantization parameters to each out-channel. With this setting, the MAC operation in Eq. 21 can be represented as:

$$\hat{A}_n = \sum_m (\Delta_{w_n} \hat{W}_{n,m}^{int})(\Delta_x \hat{x}_m^{int}) + b_n$$

$$= \Delta_{w_n} \Delta_x \sum_m \hat{W}_{n,m}^{int} \hat{x}_m^{int} + b_n \tag{22}$$

where $\Delta_{w_n}$ is scale factor for the $n_{th}$ out-channel of weights. However, the channel-wise quantization for activations sets quantization parameters to each in-channel. This setting is hardly supported

by hardware, as the MAC operation is performed as follows:

$$\hat{A}_n = \sum_m (\Delta_w \hat{W}_{n,m}^{int})(\Delta_{x_m} \hat{x}_m^{int}) + b_n$$

$$= \Delta_w \sum_m \Delta_{x_m} \hat{W}_{n,m}^{int} \hat{x}_m^{int} + b_n \tag{23}$$

where $\Delta_{x_m}$ is scale factor for the $m_{th}$ in-channel of activations. Due to its inability to factor out the different scales from the accumulator summation, it is not hardware-friendly, leading to invalid acceleration.

## I   HUMAN PREFERENCE EVALUATION

In this section, we use an open-source *aesthetic predictor*[9] to evaluate Aesthetic Score ↑, mimicking human preference assessment of the generated images. As reported in Table 10, DilateQuant has a better aesthetic representation compared to EfficientDM, which demonstrates that the quantized models with our method are more aesthetically pleasing to humans. For the large text-to-image model, we use the convincing DrawBench benchmark to evaluate human performance, as shown in Figure 6. Additionally, we visualize the random samples of quantization results in Figure 7 (LSUN-church), 8 (LSUN-Bedroom), and 9 (ImageNet). As can be seen, DilateQuant outperforms previous methods in terms of image quality, fidelity, and diversity.

Table 10: Aesthetic assessment of the different quantized models with 4-bit quantization.

| Method | LSUN-Bedroom | ImageNet | DrawBench |
|---|---|---|---|
| FP | 5.91 | 5.32 | 5.80 |
| EfficientDM | 5.47 | 3.51 | 2.84 |
| DilateQuant | 5.72 | 4.85 | 5.23 |

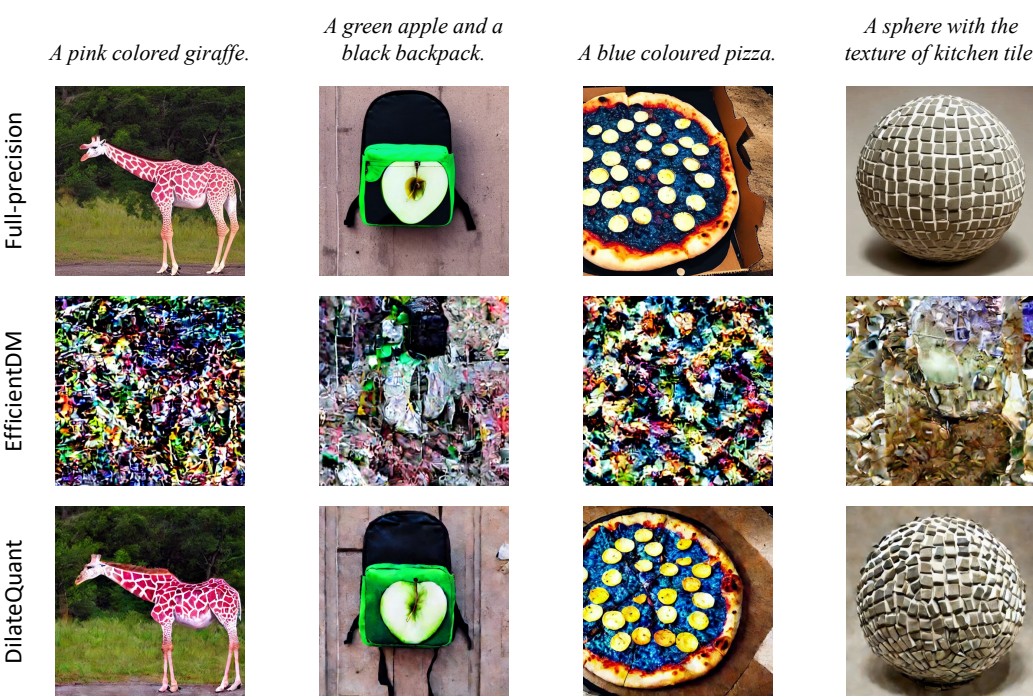

Figure 6: Random samples of different quantized models on DrawBench with 6-bit quantization.

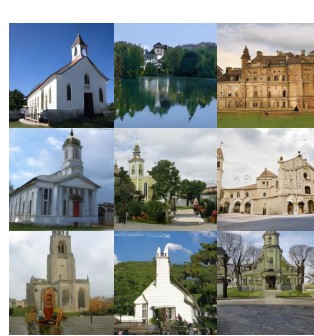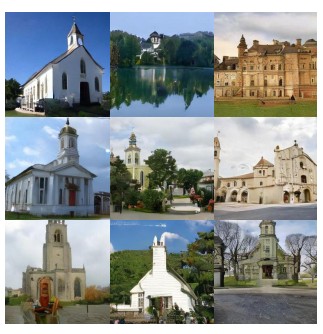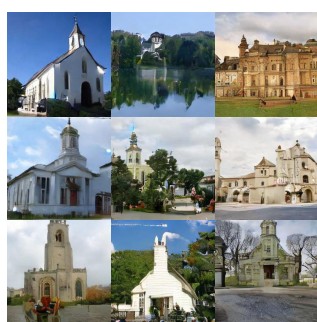

| Full-precision(W32A32) | DilateQuant(W6A6) | DilateQuant(W4A4) |

Figure 7: Random samples of quantized models with DilateQuant on LSUN-Church.

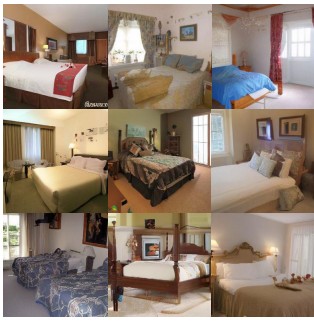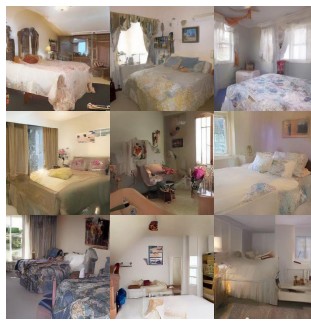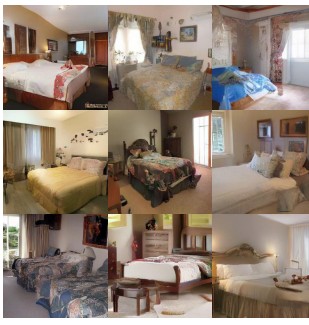

| Full-precision(W32A32) | EfficientDM(W4A4) | DilateQuant(W4A4) |

Figure 8: Random samples of different quantized models on LSUN-Bedroom with 4-bit quantization.

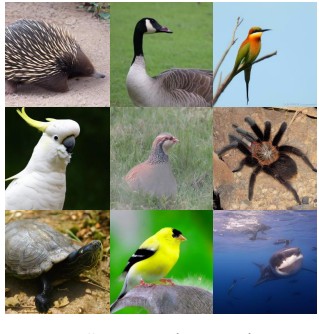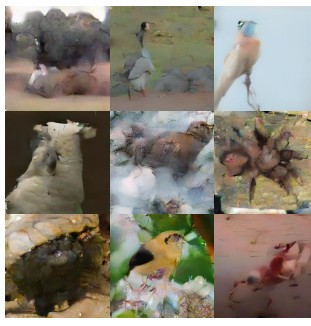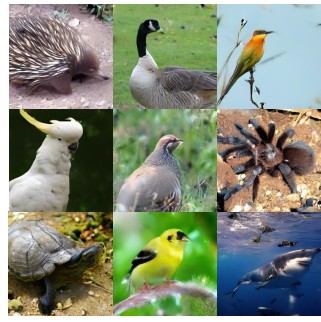

| Full-precision(W32A32) | EfficientDM(W4A4) | DilateQuant(W4A4) |

Figure 9: Random samples of different quantized models on ImageNet with 4-bit quantization.

