# OpenReview forum: "DilateQuant: Accurate and Efficient Diffusion Quantization via Weight Dilation"
_ICLR.cc/2025/Conference — Submitted to ICLR 2025_

### Official Review · Reviewer_umqh · 2024-10-31

**Soundness:** 3
**Presentation:** 3
**Contribution:** 2
**Rating:** 3
**Confidence:** 4

**Summary:**

The paper introduces DilateQuant, a quantization framework designed for diffusion models that enhances efficiency while preserving accuracy. It consists of three key components: Weight Dilation (WD), Temporal Parallel Quantizer (TPQ), and Block-wise Knowledge Distillation (BKD). WD scales activations to be more quantization-friendly without introducing additional quantization difficulty to weights. TPQ adapts to time-varying activations by setting specific quantization parameters for different temporal stages. BKD employs a block-level distillation strategy, transferring knowledge from a full-precision model to its quantized version, which reduces the backpropagation path and memory demands. Comprehensive experiments demonstrate that DilateQuant surpasses existing methods in reducing quantization errors and improving efficiency across diverse models and configurations.

**Strengths:**

The paper is written with exceptional clarity. The motivation is well-articulated and compelling. The proposed weight dilation introduces a novel technique for adjusting in-channel weights to minimize activation range while avoiding added complexities in weight quantization.

**Weaknesses:**

1. The proposed weight dilation method successfully maintains the max-min values of each out-channel weight unchanged, thereby avoiding additional quantization difficulties. However, there are potential limitations to consider. For example, if the index of an activation outlier coincides with the in-channel index corresponding to these max-min values, the method might struggle to effectively manage these outliers. In such scenarios, the proposed technique could encounter challenges in achieving its intended quantization effectiveness. Could the authors provide an analysis of how frequently activation outliers coincide with max-min weight indices in diffusion models?

2. The constraint in the proposed weight dilation method, which aims to maintain the max-min values of each out-channel weight unchanged, might be overly restrictive. A slight adjustment of these max-min values could potentially introduce minimal quantization errors while providing a more flexible approach to manage the issues noted in the previous point. This adjustment could enable the method to handle cases where the activation outlier index coincides with the in-channel index of max-min values. For instance, considering both the activation quantization error and weight error simultaneously could help determine the most effective channel-wise scaling.

3. The proposed weight dilation method calculates channel-wise scaling based solely on the magnitude of weights, without considering activations. This approach might not sufficiently mitigate extreme activation outliers. A more comprehensive strategy that takes into account both the magnitudes of activations and weights could potentially offer a more balanced and effective solution for managing these outliers. Could the authors provide a comparison of the proposed weight dilation with existing methods [1][2], which jointly consider the magnitudes of activations and weights to determine channel-wise scaling?

4. The novelty and contributions of the proposed Temporal Parallel Quantizer (TPQ) and Block-wise Knowledge Distillation (BKD) appear to be limited. Specifically, the TPQ’s methodology of assigning unique quantization parameters for each time step has already been addressed in prior works [3][4]. Furthermore, the use of block-wise reconstruction in BKD, although effective, follows a well-known path in the post-training quantization (PTQ) of diffusion models as in [3], [5], and [6]. The primary distinction lies in BKD’s approach: instead of relying solely on calibration, it updates weight parameters and quantization parameters through gradient-based methods.

Reference:

[1] SmoothQuant: Accurate and Efficient Post-Training Quantization for Large Language Models. ICML 2023.

[2] Outlier Suppression: Pushing the Limit of Low-bit Transformer Language Models. NeurIPS 2022.

[3] TFMQ-DM: Temporal Feature Maintenance Quantization for Diffusion Models. CVPR 2024.

[4] EfficientDM: Efficient Quantization-Aware Fine-Tuning of Low-Bit Diffusion Models. ICLR 2024.

[5] Post-training Quantization on Diffusion Models. CVPR 2023.

[6] Q-Diffusion: Quantizing Diffusion Models. ICCV 2023.

**Questions:**

1. The formulations presented in Eqs. (10) and (11) may encounter issues if the denominator becomes zero, leading to undefined results. A safeguard or modification in these equations to handle zero denominators is necessary to ensure their correctness and stability under all conditions. For instance, adding a small epsilon value to the denominator could prevent such undefined outcomes.

2. The comparison of calibration sample numbers between DilateQuant and EfficientDM in Tables 1 and 2 seems inaccurate. EfficientDM operates as a data-free method that does not depend on traditional training data; instead, it employs random Gaussian noise to generate images for network-wide reconstruction. Therefore, the number of calibration samples for EfficientDM should be indicated as zero. It’s possible the authors intended to reference the number of training steps instead.

3. The ablation studies presented are inadequate. For instance, the performance impact of using Weight Dilation (WD) alone (without TPQ and BKD) is not clearly isolated in Table 3. It would be beneficial to provide specific results demonstrating the effect of implementing WD independently to better understand its individual contribution to the overall performance enhancements.

---

> ### Author Response · Authors · 2024-11-22
>
> We thank the reviewer for the positive review and constructive comments. We provide our responses as follows.
>
> >**Q1: If the index of an activation outlier coincides with the in-channel index corresponding to these max-min values, the method might struggle to effectively manage these outliers. Could the authors provide an analysis of how frequently activation outliers coincide with max-min weight indices in diffusion models?**
>
> **A1**: Thank you for your valuable question. WD performs scaling based on weight range and is indeed unable to manage outliers associated with saturated weight channels. Fortunately, **we find that the extreme outliers often correspond to the unsaturated weight channels.** This distribution is further supported by Study [11]. WD aims to eliminate more extreme outliers in the activation channels while maintaining the weight range. This results in a more uniform and smaller range of activations, thereby reducing the difficulty of activation quantization. Please refer to Figure 1 (a) and (b) in the overall rebuttal, most of the extreme outliers (values greater than 100) are effectively eliminated after WD. To further demonstrate the effectiveness of WD in managing these outliers, as mentioned in "Appendix F'' of the paper, we showcase the effectiveness of WD using $E_{clip}$ metric and ***proportion of $s > 1$*** metric. The $E_{clip}$ is caused by extreme outliers, as illustrated in "Sec 3.1'' of the paper. Scaling of over 30% of the channels and significantly reducing $E_{clip}$ demonstrate WD's capability in managing extreme outliers.
> >**Q2: Considering both the activation and weight range may help determine the most effective channel-wise scaling. Could the authors provide a comparison of the proposed WD with existing methods[1][2], which jointly consider the magnitudes of activations and weights to determine channel-wise scaling?**
>
> **A2**: By considering your valuable suggestion, we replied your concern in overall rebuttal (1). Previous scaling methods consider both the magnitudes of activations and weights to determine channel-wise scaling for LLMs. However, we find that these methods are not suitable for DMs, **as outliers in DMs are distributed across all channels, whereas in LLMs, they are distributed only in certain channels.** These methods are effective for LLMs because scaling has minimal impact on the weight range and does not interfere with the PTQ process. When these methods are applied to DMs, the weight range increases significantly. Since weights and activations are multiplicatively related, smaller activation quantization errors are amplified by larger weights, resulting in no errors reduction. The same applies to weight quantization errors. Furthermore, the excessive disruption of the initial weight range hinders the model's convergence during the QAT training process. In contrast, **by carefully identifying and scaling unsaturated weight channels, WD steadily reduces the activation quantization errors while maintaining the weight range.** In "Appendix G'' of the paper, we provide a detailed comparison of the proposed WD with existing methods, in which channel-wise scalings are determined by both the activation and weight range. The results demonstrate that our approach significantly outperforms these methods for DMs. Following your suggestion, we have added a comparison between WD and OS+[10](the advanced OS[2]), and we present all comparison results in Table.
>
> | Methods | No scaling | SmoothQuant[1] | OmniQuant[7] | AWQ[8] | DGQ[9] | OS+[10] | Ours |
> |:---------:|:------------:|:----------------:|:--------------:|:--------:|:--------:|:--------:|:------|
> | proportion of $s > 1$ | 0% | 100% | 100% | 1% | 0.5% | 100% | 39.2% |
> | FID$\downarrow$ | 9.63 | 9.99 | 9.86 | 10.34 | 9.72 | 9.78 | **9.13** |
> | sFID$\downarrow$ | 7.08 | 7.29 | 7.34 | 7.53 | 7.78 | 7.23 | **6.92** |
> | IS$\uparrow$ | 8.45 | 8.46 | 8.50 | 8.38 | 8.52 | 8.36 | **8.56** |
>
> Reference:
> [1] SmoothQuant: Accurate and Efficient Post-Training Quantization for Large Language Models. ICML 2023.
> [2] Outlier Suppression: Pushing the Limit of Low-bit Transformer Language Models. NeurIPS 2022.
> [3] TFMQ-DM: Temporal Feature Maintenance Quantization for Diffusion Models. CVPR 2024.
> [4] EfficientDM: Efficient Quantization-Aware Fine-Tuning of Low-Bit Diffusion Models. ICLR 2024.
> [5] Post-training Quantization on Diffusion Models. CVPR 2023.
> [6] Q-Diffusion: Quantizing Diffusion Models. ICCV 2023.
> [7] Omniquant: Omnidirectionally calibrated quantization for large language models. ICLR 2024.
> [8] AWQ: Activation-aware Weight Quantization for LLM Compression and Acceleration. MLSys 2024.
> [9] Dual Grained Quantization: Efficient Fine-Grained Quantization for LLM.
> [10] Outlier Suppression+ : Accurate quantization of large language models by equivalent and effective shifting and scaling. EMNLP 2023.
> [11] PTQ4DiT: Post-training Quantization for Diffusion Transformers. NeurIPS 2024.

---

> ### Author Response · Authors · 2024-11-22
>
> >**Q3: The novelty and contributions of the proposed TPQ and BKD appear to be limited. The TPQ’s methodology has been addressed in prior works[3,4] and the BKD follows block-wise reconstruction in PTQ methods[3,5,6].**
>
> **A3**: By considering your valuable suggestion, we replied your concern in overall rebuttal (2). TPQ and BKD differ significantly from previous methods. Previous approaches set multiple quantization parameters by stacking multiple quantizers. These independent quantizers require step-by-step retraining across each timestep, leading to a significant demand for calibration samples and training time. Conversely, **TPQ sets time-step quantization parameters in a single quantizer, supporting parallel optimization through an indexing approach.** This significantly reduces the need for calibration samples and training time. On the other hand, reconstruction-based training methods in PTQ optimize quantization parameters. These approaches fail to meet the accuracy demand of low-bit quantization. The QAT-based training approaches retrain the entire model’s weights within a single backpropagation step. These approaches suffer from training instability and require extensive training time and memory resources. In contrast, **BKD jointly optimizes quantized weights and quantization parameters with a short backward path, which significantly enhance training stability and reduce both training time and memory consumption.**
> **The contributions of the proposed TPQ and BKD lie in addressing the issues of low accuracy in PTQ methods and low efficiency in QAT methods.**
> By employing TPQ and BKD, our approach achieves QAT-like accuracy with PTQ-like efficiency. As shown in Table, we compare our method with the existing works you mentioned to highlight our advantages in both accuracy and efficiency.
>
> |Task|Methods|Framework|FID$\downarrow$|sFID$\downarrow$|IS$\uparrow$|Calib.|Time Cost|GPU Memory|
> |:-----:|:-----:|:-----:|:-----:|:-----:|:-----:|:-----:|:-----:|:-----|
> |ImageNet|PTQ4DM[5]|PTQ|263.27|162.59|2.55|1024|6.43 h|13810 MB|
> | |Q-Diffusion[6]|PTQ|260.23|161.48|2.42|1024|6.43 h|13831 MB|
> | |TMFQ-DM[3]|PTQ|258.81|152.42|2.40|10240|7.53 h|14682 MB|
> | |EfficientDM[4]|QAT|12.08|14.75|122.12|102.4K|5.20 h|22746 MB|
> | |**Ours**|QAT|**8.01**|**13.92**|**257.24**|1024|6.56 h|14680 MB|
> |CIFAR-10|EfficientDM[4]|QAT|81.27|30.95|6.68|1.63M|2.98 h|9546 MB|
> | |**Ours**|QAT|**9.13**|**6.92**|**8.56**|5120|1.08 h|3439 MB|
> |MSCOCO|EfficientDM[4]|QAT|216.43|111.76|14.35|12.8K|30.25 h|46082 MB|
> | |**Ours**|QAT|**44.82**|**42.97**|**23.51**|512|7.41 h|31942 MB|
>
> >**Q4: The formulations presented in Eqs. (10) and (11) may encounter issues if the denominator becomes zero.**
>
> **A4**: Thanks for your correction. In code, we use $.clamp(min=1e-5)$ and $.clamp(max=-1e-5)$ separately to avoid division by zero in Eqs. (10) and (11). For brevity in the paper, we simplified them to $.clamp(min=0)$ and $.clamp(max=0)$. In the final version, we will revise the equations to:
> $s_{k1} = min(W_{max}/W_k.clamp(min=1e-5))$
> $s_{k2} = min(W_{min}/W_k.clamp(max=-1e-5))$
>
> >**Q5: The comparison of calibration sample numbers between DilateQuant and EfficientDM in Tables 1 and 2 seems inaccurate. The number of calibration samples for EfficientDM should be indicated as zero.**
>
> **A5**: Most of quantization methods for diffusion models are data-free, meaning they do not rely on the original training dataset. **However, all of methods still require pre-trained models to generate calibration samples before quantization for either calibration or training purposes.** Thus, calibration samples are also considered a type of training resource. For example, the latest work, TFMQ-DM[3], requires running a pre-trained model for **0.4 hours** to obtain **10240** calibration samples.
> We report the number of calibration samples required by different methods to illustrate their resource consumption. "Table 4" in the paper presents the requirements of different methods for calibration samples (Calib.) and the original training dataset (Training Data). EfficientDM does not require the original training dataset. However, **it demands hundreds of thousands of calibration samples.**
>
> >**Q6: The ablation studies presented are inadequate. It would be beneficial to provide specific results demonstrating the effect of implementing WD independently.**
>
> **A6**: Thank you for your valuable suggestion. We have detailed the contribution of each component to performance improvement in overall rebuttal (3). The proposed WD significantly improves the FID score from **120.24** to **26.26** within the PTQ framework and from **18.45** to **9.66** within the QAT framework, demonstrating its effectiveness.
>
> ---
> Thank you again for your valuable comments and hope our response can resolve your concerns. Please let us know if you have any further questions. If you feel your concerns are addressed, please consider reevaluating our work. Looking forward to hearing from you.

---

> ### Author Response · Authors · 2024-11-25
> **Kindly Inquiry to Reviewer umqh**
>
> Dear Reviewer umqh,
>
> We greatly appreciate the time and effort in reviewing our work. We have carefully considered your comments and suggestions and have made significant revisions to address the concerns you raised. We are eager to ensure that our paper meets the high standards of our respected reviewers.
>
> We understand your busy schedule, but given that the author-reviewer discussion stage is coming to an end, we sincerely hope that your concerns can be addressed promptly. Please don’t hesitate to let us know if there is any additional feedback you might have at this stage.
>
> Best regards,
> Authors of #2486.

---

> > ### Comment · Reviewer_umqh · 2024-11-26
> > **Official Comment by Reviewer umqh**
> >
> > Thanks to the authors for the rebuttal and revisions. While I acknowledge the efforts made, my concerns remain unresolved after reviewing the updated version.
> >
> > First, based on Figure 1 in the provided PDF, it appears that activation outliers occur only in certain channels. The proposed method scales only a subset of these channels; however, the prior work OS+ seems to adopt a similar strategy by selectively scaling channels—specifically those whose maximum values exceed a threshold $t$ —rather than uniformly scaling all channels.
> >
> > Second, the distinction between the proposed TPQ and existing methods remains unclear. Both approaches use a single quantizer, assign unique quantization parameters to each time step, and support parallel training. Moreover, the use of layer-by-layer distillation is not novel; for example, ZeroQuant [7] applies this technique in large language models (LLMs), employing a similar approach to the proposed method by “jointly optimizing quantized weights and quantization parameters with a short backward path.” Providing further clarification on the methodological or practical advancements that TPQ offers over these prior works would help to better highlight its contribution.
> >
> > **Reference:**
> >
> > [7] ZeroQuant: Efficient and Affordable Post-Training Quantization for Large-Scale Transformers. NeurIPS 2022.

---

> ### Author Response · Authors · 2024-11-27
>
> Dear umqh,
>
> We sincerely appreciate your positive reply. After carefully considering your concerns, we provide our responses as follows.
> >**Q1: The activation outliers occur only in certain channels. The proposed method scales only a subset of these channels. The prior work OS+ seems to adopt a similar strategy by selectively scaling certain channels with maximum values.**
>
> **A1**: Thank you for your valuable comment. Considering that the top **75%** of absolute activation values in each channel of DMs account for only **0.1%** of the total, we suggest that extreme outliers are pervasive across all channels. Please refer to Figure 2(b) of the paper, where we visualize the stark contrast in outlier distributions between LLMs and DMs. Specifically, the outliers in all channels of DMs leads to a significantly broader range of activation magnitudes compared to LLMs. **Our approach scales as many activation channels as possible while preserving the weight range, as evidence by more over 30% of the scaling channels (*proportion of s>1*) and approximately 0.9× reduction in activation range ($\Delta^{‘}/\\Delta$) in Table 8 of the paper.** As shown in Figure 2(a)(b) of the overall response PDF, we analyze the average magnitude of activations across all channels before and after scaling, observing a reduction from **51.23** to **45.72**.
>
> **The previous OS+ method applies shifting and scaling to all channels to reduce the quantization errors in LLMs.** Similar to SmoothQuant, it significantly increases the magnitude of weights for DMs. **The previous DGQ method selectively scales the top 0.5% of activation channels with the maximum values.** Experimental results show that **this approach is insufficient to address the pervasive outliers present across all channels in DMs.** For example, DGQ only reduces the average magnitude from **51.23** to **50.95** for the activations in Figure 2(a). Additionally, it fails to fully preserve the weights range. In Appendix G of the paper, we presented a comparison between our method and DGQ. In the updated version, we have further included a comparison with OS+. Our method achieves a **0.59** and **0.65** improvement in FID score over DGQ and OS+, respectively.
> >**Q2: The distinction between the TPQ and existing methods remains unclear. Both approaches use a single quantizer and support parallel training.**
>
> **A2**: Thank you for your valuable feedback. We provide a detailed comparison between our method and existing approaches as follows. Please refer to the [QuEST code](https://github.com/hatchetProject/QuEST/blob/main/qdiff/quant_layer.py), where the *TimewiseUniformQuantizer* is defined by stacking single quantizers for each time step (lines 66–67). During inference (lines 69–83), it can only call the forward function of one quantizer at a step.
> In the [EfficientDM code](https://github.com/ThisisBillhe/EfficientDM/blob/main/quant_scripts/quant_layer.py) (lines 82-96), the *self.current_step* parameter is consistently a scalar. For an multiple time-step input, it cannot apply the corresponding quantization parameters for each time step, thus lacking the capability for parallel training.
> TFMQ-DM employs a dynamic inference strategy to preform time-step quantization parameters. This capability arises entirely from the inherent advantages of real-time computation in dynamic inference. Consequently, in [TFMQ-DM code](https://github.com/ModelTC/TFMQ-DM/blob/main/quant/quant_layer.py) (lines 211-227), it does not assign separate quantization parameters for different time steps.
> Furthermore, the training codes for these methods reveal that they require a time-step-wise loop to individually train the quantization parameters.
>
> To provide a clearer understanding of the unique design of our TPQ and its capability to support parallel training, we have included DilateQuant code in the supplementary materials.
> Please refer to *./DilateQuant/quant/quant_layer.py* (lines 209-256). The parameter *self.is_mix_steps* determines whether the quantizer follows a parallel inference path or a step-by-step inference path. During parallel inference, the quantizer uses an indexed tensor *self.t* to retrieve the corresponding quantization parameters for the time-step input. This design enables simultaneous training of multiple quantization parameters, as the code *./DilateQuant/quant/block_w_recon.py* (lines 195, 227-230).
> >**Q3: The use of layer-by-layer distillation is not novel. ZeroQuant applies the similar approach.**
>
> **A3**: **BKD differs from ZeroQuant in terms of training framework and target.** ZeroQuant performs layer-by-layer distillation within a PTQ framework, focusing solely on training quantization parameters. In contrast, BKD adopts a QAT framework with block-by-block joint optimization of both quantized weights and quantization parameters.
>
> ---
> we sincerely hope that your concerns can be addressed. Please don’t hesitate to let us know if there is any additional feedback you might have.

---

> > ### Comment · Reviewer_umqh · 2024-12-02
> > **Official Comment by Reviewer umqh**
> >
> > I appreciate the additional clarification. However, my concerns remain. The distinction between TPQ and prior methods seems to be largely one of implementation detail. All methods employ a single quantizer, albeit with different quantization parameters for different time steps, which allows for potential parallel training.

---

> ### Author Response · Authors · 2024-12-01
> **Further Inquiry to Reviewer umqh**
>
> Dear Reviewer umqh,
>
> We sincerely apologize for disturbing you again. We are truly grateful for the time and effort you have dedicated to reviewing our work, and for carefully considering our responses. Your constructive comments and feedback have made a significant contribution to improving our work.
> Based on your suggestions, we have made the following improvements in our new submission:
> - We have added theoretical proof demonstrating the superiority and novelty of WD over previous scaling algorithms, supported by experimental data;
> - We have provided code to showcase the differences between TPQ, BKD, and prior methods, and have clarified the descriptions of TPQ and BKD in the paper;
> - We have included more comprehensive ablation experiments to validate the effectiveness of the individual components of our method.
>
> With the sincere intention of addressing all the reviewer’s concerns, we would like to kindly confirm whether there are any remaining issues with our work. We will promptly respond to any feedback you may have before the author-reviewer discussion phase concludes, so please feel free to express any questions. If you feel your concerns are addressed, please consider reevaluating our work. Once again, thank you for your thoughtful review, and we look forward to your reply.
>
> Best regards,
> Authors of #2486.

---

> ### Author Response · Authors · 2024-12-02
> **Authors Response to Reviewer umqh**
>
> Dear Reviewer umqh,
>
> We appreciate your response. By carefully considering your concerns, we would like to clarify the differences between the proposed TPQ and previous methods as follows:
>
> Currently, methods that set multiple quantization parameters for a single layer include QuEST, TFMQ-DM, and EfficientDM. We will  illustrate the distinctions between these methods and TPQ in terms of quantizer objects, quantizer settings, and quantizer targets.
>
> (1). In terms of quantizer objects, QuEST, EfficientDM, and TPQ all set multiple quantization parameters for all layers, **while TFMQ-DM only sets multiple quantization parameters for the embedding layers and time embeddings.** This difference is one of the key factors contributing to the significantly better performance of our method (FID score↓ 8.01) compared to TFMQ-DM (FID score↓ 258.81), as shown in Table of A3 of our initial response.
>
> (2). In terms of quantizer settings, **QuEST implements multiple quantization parameters by using multiple quantizers for a single layer.** It reduces the number of quantizers used by grouping them along the time steps. For example, in a model with 100 time steps, QuEST sets 10 quantizers for a single layer, with the first quantizer applied to the first 10 time steps. **EfficientDM retrieves quantization parameters corresponding to specific time steps by cumulating and resetting *self.current_step*.** This parameter is a scalar that is only related to the number of model iterations, with **no relation** to the temporal sequence of input samples, thus not supporting parallel quantization. **In sharp contrast, TPQ uses an indexing approach that maps the selection of quantization parameters to the time steps of the input samples, enabling both independent inference during sampling and parallel inference during training.** The specific implementation code for these methods has been provided in A2 of our second response.
>
> (3). In terms of quantizer targets, the target of previous methods using multiple quantization parameters is to reduce quantization errors. While these quantizers have indeed enhanced model accuracy, **they rely on the expensive independent optimization of each quantization parameter, significantly reducing quantization efficiency.** Specifically, these methods require extensive loops to optimize the quantization parameters for each time step, as shown in training code of [QuEST](https://github.com/hatchetProject/QuEST/blob/main/qdiff/post_layer_recon_uncond.py) (lines 57-58, 192) and [EfficientDM](https://github.com/ThisisBillhe/EfficientDM/blob/main/ldm/models/diffusion/ddim.py) (lines 338, 389-403). **A key distinction of TPQ, however, is its ability to enhance model accuracy while maintaining high efficiency. This aligns with the title of our paper, "Accurate and Efficient Diffusion Quantization".** Ablation studies and task-specific experiments demonstrate that, compared to other methods, TPQ not only improves model accuracy to the level of QAT but also maintains quantization efficiency at the level of PTQ.
>
> Best regards,
> Authors of #2486.

---

### Official Review · Reviewer_8d1A · 2024-11-01

**Soundness:** 3
**Presentation:** 3
**Contribution:** 2
**Rating:** 3
**Confidence:** 4

**Summary:**

Quantization is one of the most important method to reduce the model size and improve the computing efficiency. This manuscript proposes a novel quantization framework (DilateQuant ) for diffusion models, which offers comparable accuracy and high efficiency. The main contribution of the manuscript is that it introduces Weight Dilation (WD) that costlessly absorbs the activation quantization errors into weight quantization. In addition, a Temporal Parallel Quantizer (TPQ) and a Block-wise Knowledge Distillation (BKD) are further proposed to improve the quantization quality. However, the main issue is that this paper innovation. It combines a few existing methods to achieve a good quantization effect.

**Strengths:**

The most innovative point of this paper is the proposed Weight Dilation (WD) technology, which maximally expands the weight in the unsaturated channel to the restricted range by equivalent scaling mathematically. WD can absorb activation quantization error into weight quantization with no extra cost. The activation range is reduced, making it easy to quantify the activation. The weight range remains the same, which makes it easy for the model to converge during the training phase. DilateQuant is significantly superior to existing methods in terms of accuracy and efficiency.

**Weaknesses:**

1. It can be seen from Table 3 that the biggest innovation in this paper, WD, has little improvement on performance.
2. The full text lacks innovation. For example, TPQ and BKD in this paper are not innovative.
3. Setting the scale factor for the out-channel of weights is not a new method, which has been widely used in the channel-wise quantization.
4. The paper lacks the effect of image generation at higher resolution (i.e., 1024x1024).

**Questions:**

1. Please add the results of delta'x/delta x in Table 8
2. This work is strongly correlated with other equivalent scaling algorithms. Please add the results of W6A6 in Table 9 to demonstrate the generalization ability.
3. Please explain the difference between TPQ in this article and TLSQ in EfficientDM[1]. It seems that the training process is the only difference that each time step has different activation quantization parameters.
4. Please explain the block splitting method of BKD. In addition, please explain the difference between BKD and the related work BK-SDM[2].
5. If possible, please supplement the comparison results with PTQD[3] and TFMQ-DM[4].

[1] He Y, Liu J, Wu W, et al. Efficientdm: Efficient quantization-aware fine-tuning of low-bit diffusion models[J]. arXiv preprint arXiv:2310.03270, 2023.
[2] Kim B K, Song H K, Castells T, et al. Bk-sdm: A lightweight, fast, and cheap version of stable diffusion[J]. arXiv preprint arXiv:2305.15798, 2023.
[3] He Y, Liu L, Liu J, et al. Ptqd: Accurate post-training quantization for diffusion models[J]. Advances in Neural Information Processing Systems, 2024, 36.
[4] Huang Y, Gong R, Liu J, et al. Tfmq-dm: Temporal feature maintenance quantization for diffusion models[C]//Proceedings of the IEEE/CVF Conference on Computer Vision and Pattern Recognition. 2024: 7362-7371.

---

> ### Author Response · Authors · 2024-11-22
>
> We thank the reviewer for the positive review and constructive comments. We provide our responses as follows.
>
> >**Q1: The biggest innovation in this paper, WD, has little improvement on performance.**
>
> **A1**: We apologize for the unclear ablation experiments and reply to your concern in overall rebuttal (3). The proposed WD significantly improves the FID score↓ from **120.24** to **26.26** within the PTQ framework and from **18.45** to **9.66** within the QAT framework. Compared to the latest method (TMFQ-DM [1]) with the PTQ framework, which achieved an FID score of **258.81**, WD significantly pushed the FID score limits of the PTQ method to **26.26**.
> >**Q2: The TPQ and BKD in this paper are not innovative. Please explain the difference between TPQ in this article and TLSQ in EfficientDM. Please explain the difference between BKD and the related work BK-SDM.**
>
> **A2**: By considering your valuable comments, we replied to your concern in overall rebuttal (2). The primary difference between TPQ and TLSQ lies in the design of the quantizer. TLSQ employs multiple quantization parameters by stacking multiple quantizers.
> These independent quantizers require step-by-step retraining across each timestep, leading to a significant demand for calibration samples and training time. Conversely, **TPQ sets time-step quantization parameters in a single quantizer, supporting parallel optimization through an indexing approach**. This significantly reduces the need for calibration samples and training time.
> Empirically, TPQ achieves a **160×** reduction in calibration samples and a **2×** reduction in training time compared to TLSQ.
> **BKD and BK-SDM are fundamentally different methods: BKD belongs to model quantization, while BK-SDM belongs to model pruning.** BKD focuses on block-level distillation to mitigate errors introduced by quantization, while BK-SDM performs distillation on the entire model to remove redundant layers.
> >**Q3: Setting the scale factor for the out-channel of weights has been widely used in the channel-wise quantization.**
>
> **A3**: **We replied your concern in overall rebuttal (1).** Previous scaling methods were designed for LLMs. However, due to the significantly different outlier distributions in LLMs and DMs, these methods are not suitable for DMs. Our scaling method is specifically designed based on the unique activation-weight distributions of DMs. By carefully identifying and scaling unsaturated weight channels, WD reduces the activation range without altering the weight range. This results in a consistent reduction in quantization difficulty and improved model stability during training.
> >**Q4: The paper lacks the effect of image generation at higher resolution (i.e., 1024x1024).**
>
> **A4**: Most of the quantization methods are evaluated on Stable Diffusion (512×512) to assess their effectiveness in generating high-resolution images. As mentioned in "Table 2'', "Table 10'', and "Figure 3'' of the paper, our method achieves usable image quality and semantic alignment on large text-to-image models, outperforming previous approaches. In the future, we will extend our method to higher-resolution models.
>
> Reference:
> [1] TFMQ-DM: Temporal Feature Maintenance Quantization for Diffusion Models. CVPR 2024.

---

> ### Author Response · Authors · 2024-11-22
>
> >**Q5: Please add the results of Δx′/Δx in Table 8 and add the results of W6A6 in Table 9 to demonstrate the generalization ability.**
>
> **A5**: Thank you for your valuable suggestion. In "Table 8" of the paper, instead of illustrating changes in Δx, we present more direct metrics, **$E_{clip}$** and ***proportion of s > 1***, which are influenced by Δx, to demonstrate the impact of WD on the quantization of activation *x*. Following your suggestion, we add Δx′/Δx to Table. All metrics demonstrate that WD effectively alleviates the difficulty of activation quantization while maintaining weight ranges.
>
> | Metrics                   | CIFAR-10 | LSUN-Bedroom | LSUN-Church | ImageNet | MSCOCO |
> |---------------------------|:--------------:|:--------------:|:--------------:|:--------------:|:--------------:|
> | Δw′ / Δw | 1.02 | 1.02 | 1.01 | 1.01| 1.02 |
> | $E_{clip}$ | 0.83 | 0.92 | 0.92 | 0.93 | 0.92 |
> | proportion of s > 1 | 39.2% | 52.4% | 32.8% | 36.5%    | 43.8%  |
> | Δx′ / Δx | 0.91| 0.92| 0.91 | 0.92 | 0.90|
>
> We have included the W6A6 results to "Table 9''. The results in the updated Table demonstrate that WD outperforms existing scaling methods across various bit-widths.
>
> | Metrics| Prec. | No scaling | SmoothQuant | OmniQuant | AWQ   | DGQ   | Ours |
> |:-----------------------:|:------------:|:------------:|:------------:|:------------:|:------------:|:------------:|:------------|
> | proportion of s > 1|W4A4| 0% | 100% | 100% | 1%| 0.5%  | 39.2% |
> | FID ↓ |W4A4| 9.63 | 9.99 | 9.86| 10.34 | 9.72  | **9.13**       |
> | sFID ↓ |W4A4 | 7.08| 7.29| 7.34| 7.53  | 7.78  | **6.92**       |
> | IS ↑|W4A4| 8.45| 8.46 | 8.50| 8.38  | 8.52  | **8.56**       |
> | FID ↓| W6A6 | 5.75| 5.44| 5.56 | 5.85  | 5.09  | **4.46** |
> | sFID ↓| W6A6 | 4.96 | 4.87 | 4.89 | 5.19  | 4.84  | **4.64** |
> | IS ↑| W6A6 | 8.80 | 8.86 | 8.81 | 8.78  | 8.89  | **8.92** |
>
> >**Q6: Please supplement the comparison results with PTQD[3] and TFMQ-DM[4].**
>
> **A6**: Following your suggestion, we present a detailed comparison between our method, PTQD, and TFMQ-DM in Table. When the bit precision is reduced to 4-bit, both PTQD and TFMQ-DM experience significant performance degradation. In contrast, our method effectively recovers the accuracy of models. On the other hand, our method maintains efficiency comparable to TFMQ-DM and PTQD.
>
> | Method | Prec. (W/A) | FID $\downarrow$ | sFID $\downarrow$ | IS $\uparrow$ | Calib. | Time Cost | GPU Memory |
> |:--------:|:-------------:|:-------------:|:-------------:|:-------------:|:-------------:|:-------------:|:-------------|
> | PTQD | 6/6 | 16.38 | 17.79 | 146.78 | 1024 | 4.63 h | 13836 MB |
> | TFMQ-DM | 6/6 | **7.83** | 8.23 | 311.32 | 10240 | 7.53 h | 14682 MB |
> | Ours | 6/6 | 8.25 | **7.66** | **312.30** | 1024 | 6.56 h | 14680 MB |
> | PTQD | 4/4 | 245.84 | 107.63 | 2.88 | 1024 | 4.63 h | 13836 MB |
> | TFMQ-DM | 4/4 | 258.81 | 152.42 | 2.40 | 10240 | 7.53 h | 14682 MB |
> | Ours | 4/4 | **8.01** | **13.92** | **257.24** | 1024 | 6.56 h | 14680 MB |
>
> ---
> Thank you again for helping us improve the paper and hope our response can resolve your concerns. Please let us know if you have any further questions. We will be actively available until the end of rebuttal period. If you feel your concerns are addressed, please consider reevaluating our work. Looking forward to hearing from you.

---

> ### Author Response · Authors · 2024-11-25
> **Kindly Inquiry to Reviewer 8d1A**
>
> Dear Reviewer 8d1A,
>
> We greatly appreciate the time and effort in reviewing our work. We have carefully considered your comments and suggestions and have made significant revisions to address the concerns you raised. We are eager to ensure that our paper meets the high standards of our respected reviewers.
>
> We understand your busy schedule, but given that the author-reviewer discussion stage is coming to an end, we sincerely hope that your concerns can be addressed promptly. Please don’t hesitate to let us know if there is any additional feedback you might have at this stage.
>
> Best regards,
> Authors of #2486.

---

> ### Author Response · Authors · 2024-12-01
> **Further Inquiry to Reviewer 8d1A**
>
> Dear Reviewer 8d1A,
>
> We sincerely apologize for disturbing you again. We are truly grateful for the time and effort you have dedicated to reviewing our work. In response to your comments, we have refined our work in several aspects:
> - We have provided theoretical proofs and ablation experiments to demonstrate the effectiveness and novelty of WD compared to previous scaling methods.
> - We have elaborated on the innovations of TPQ and BKD, highlighting their significant differences from previous approaches and showing their improvements in both efficiency and accuracy through ablation studies.
> - Following your suggestion, we have expanded the experiments in Tables 8 and 9, comparing our method with PTQD and TFMQ-DM.
>
> With the sincere intention of addressing all the reviewer’s concerns, we would like to kindly confirm whether there are any remaining issues with our work. We will promptly respond to any feedback you may have before the author-reviewer discussion phase concludes, so please feel free to express any questions. If you feel your concerns are addressed, please consider reevaluating our work. Once again, thank you for your thoughtful review, and we look forward to your reply.
>
> Best regards,
> Authors of #2486.

---

### Official Review · Reviewer_1M4q · 2024-11-04

**Soundness:** 3
**Presentation:** 3
**Contribution:** 3
**Rating:** 6
**Confidence:** 4

**Summary:**

This paper proposes a QAT framework dubbed DilateQuant for diffusion models, which consists of three components, i.e, Weight Dilation (WD), Temporal Parallel Quantizer (TPQ), and Block-wise Knowledge Distillation (BKD). The core contribution of the paper is the WD, which reduces the difficulty of activation quantization via equivalent scaling like SmoothQuant, and constrains the scaling intensity based on the maximum values of the output channels.

**Strengths:**

- The idea of the proposed Weight Dilation is novel and interesting.
- The paper is well-written and easy to understand.

**Weaknesses:**

- The contributions of TPQ and BKD appear to be modest, especially for BKD, which is well explored in the previous quantization and neural architecture search literature.
- According to the ablation study (Table. 3), most of the improvement owes to knowledge distillation (BKD). Adding the WD to the BKD only improves the FID from 9.63 to 9.13. So is the key component actually the BKD?
- The paper only validates the method on Unet-based models, however, given that DiT-based diffusion models (e.g., SD3, Pixart, FLUX) have predominated, the reviewer believes doing experiments on DiT-based models can enhance the paper. Besides, there are some papers adopting the equivalent scaling for DiT quantization[1][2], the proposed method might need to be compared with them to show the superiority of the proposed equivalent scaling method (WD).
- The reviewer is confused about the motivation of WD. The paper states that outliers exist in all channels, so equivalent scaling methods of LLMs are not suitable for DMs, which is also illustrated by Figure 2b. However, since there are “outliers” in all channels, can we still call this “outlier” an outlier? It seems that the difference in activation values of DMs across input channels is not much less salient than LLMs.

[1] PTQ4DiT: Post-training Quantization for Diffusion Transformers

[2] ViDiT-Q: Efficient and Accurate Quantization of Diffusion Transformers for Image and Video Generation

**Questions:**

Please refer to the weakness.

---

> ### Author Response · Authors · 2024-11-22
>
> We thank the reviewer for the positive review and constructive comments. We provide our responses as follows.
> >**Q1: The contributions of TPQ and BKD appear to be modest.**
>
> **A1**: By considering your valuable suggestion, we replied your concern in overall rebuttal (2). **TPQ and BKD differ significantly from previous methods.**
> Existing approaches set multiple quantization parameters for multiple time steps, improving accuracy but at the significant cost of training efficiency.
> On the other hand, reconstruction-based training methods in PTQ fail to meet the accuracy demand of low-bit quantization, while QAT-based training approaches require substantial training resources.
> In contrast, our approach achieves QAT-like accuracy with PTQ-like efficiency.
> **The contributions of the proposed TPQ and BKD lie in addressing the issues of low accuracy in PTQ methods and low efficiency in QAT methods.**
> >**Q2: According to the ablation study, most of the improvement owns to knowledge distillation (BKD). So is the key component actually the BKD?**
>
> **A2**: We apologize for the unclear ablation study.
> We replied your concern in overall rebuttal (3). BKD achieves more significant performance improvements by retraining the quantized weights, transforming our method into a QAT framework.
> In this paper, we regard WD as the key component, by considering its innovation, performance improvement, and generalization ability. **It is specifically designed based on the unique activation-weight distribution of DMs.** As a plug-and-play tool for adjusting parameter distributions, **WD seamlessly adapts to both PTQ and QAT frameworks to effectively enhance performance.**
> >**Q3: The paper only validates the method on Unet-based models. The reviewer believes doing experiments on DiT-based models can enhance the paper. Besides, the proposed method might need to be compared with scaling methods [1,2] for DiT quantization to show the superiority of the proposed WD.**
>
> **A3**: Thank you for your valuable comment. Following your suggestion, we conduct experiments on DiT-XL/2 (256×256) with 50 steps. As shown in Table, **our method is applicable to DiT-based models**. Due to time constraints, we compared our method with PTQ4DiT [1] at 4-bit and 6-bit precision. The results demonstrate that our method significantly outperforms PTQ4DiT in both accuracy and efficiency. Specifically, **while PTQ4DiT fails at 4-bit precision, our approach maintains robust performance.** Additionally, our method requires only **4.63 hours** of training compared to the **14.50 hours** needed for PTQ4DiT.
>
> | Method| Prec. (W/A) | FID↓| sFID↓| IS↑| Calib. | Time Cost | GPU Memory |
> |:-----------:|:-------------:|:-------------:|:-------------:|:-------------:|:-------------:|:-------------:|:-------------:|
> | PTQ4DiT  | 6/6| 20.68|42.56|103.24|8000|14.50 h |15564 MB |
> | **Ours**| 6/6 | **15.63** | **31.58** | **157.64** | 5120   |**4.63 h**| 15686 MB|
> | PTQ4DiT| 4/4| 256.80| 140.54| 2.27| 8000| 14.50 h| 15564 MB|
> |**Ours** | 4/4|**56.83**|**54.57**| **89.66** | 5120 |**4.63 h**| 15686 MB|
>
> >**Q4: The reviewer is confused about the motivation of WD. Since there are “outliers” in all channels, can we still call this “outlier” an outlier?**
>
> **A4**: We sincerely appreciate your thorough review and valuable questions.
> Due to the distinct distribution of extreme activations in DMs compared to LLMs, previous scaling methods fail to reduce quantization difficulty in DMs.
> Moreover, the excessive disruption of weight ranges increase instability during training.
> **The motivation of WD is to eliminate extreme activations while maintaining the weight range.**
> As shown in Figure 1 of the overall rebuttal, WD decreases the activation range while keeping the weight range unchanged, effectively reducing quantization difficulty and ensuring training stability.
> We still refer to activations across all channels with extremely large magnitudes as outliers.
> **Because they constitute less than 0.1% of the total activations within each channel and significantly impact the setting of quantization parameters.**
> From an overall distribution perspective, the differences between activation channels in DMs are less salient compared to LLMs. However, **from a scaling perspective, DMs are more salient than LLMs.** Given that extreme activations are distributed across all channels in DMs, unconstrained scaling can significantly alter the ranges of weights and activations, which is not suitable for DMs (please refer to overall rebuttal (1) for detail).
>
> Reference:
> [3] DiT: Scalable diffusion models with transformers. ICCV 2023.
>
> ---
> We hope our response can resolve your concerns. Please let us know if you have any further questions. We will be actively available until the end of rebuttal period. If you feel your concerns are addressed, please consider reevaluating our work. Looking forward to hearing from you.

---

> ### Author Response · Authors · 2024-11-25
> **Kindly Inquiry to Reviewer 1M4q**
>
> Dear Reviewer 1M4q,
>
> We greatly appreciate the time and effort in reviewing our work. We have carefully considered your comments and suggestions and have made significant revisions to address the concerns you raised. We are eager to ensure that our paper meets the high standards of our respected reviewers.
>
> We understand your busy schedule, but given that the author-reviewer discussion stage is coming to an end, we sincerely hope that your concerns can be addressed promptly. Please don’t hesitate to let us know if there is any additional feedback you might have at this stage.
>
> Best regards,
> Authors of #2486.

---

> ### Author Response · Authors · 2024-12-01
> **Further Inquiry to Reviewer 1M4q**
>
> Dear Reviewer 1M4q,
>
> We sincerely apologize for disturbing you again. We are truly grateful for the time and effort you have dedicated to reviewing our work. Based on your concerns, we have made the following improvements in our work:
> - We have elaborated on the innovations of TPQ and BKD, highlighting their significant differences from previous approaches and showing their improvements in both efficiency and accuracy.
> - We have included more comprehensive ablation experiments to validate the effectiveness of the individual components of our method.
> - We have extended our method to the DiT-based diffusion models. The experimental results demonstrate that our method outperforms existing approaches specifically designed for the DiT framework.
>
> With the sincere intention of addressing all the reviewer’s concerns, we would like to kindly confirm whether there are any remaining issues with our work. We will promptly respond to any feedback you may have before the author-reviewer discussion phase concludes, so please feel free to express any questions. If you feel your concerns are addressed, please consider reevaluating our work. Once again, thank you for your thoughtful review, and we look forward to your reply.
>
> Best regards,
> Authors of #2486.

---

### Official Review · Reviewer_sQQb · 2024-11-05

**Soundness:** 3
**Presentation:** 3
**Contribution:** 3
**Rating:** 6
**Confidence:** 4

**Summary:**

The authors proposed DilateQuant framework to quantize the diffiusion model. First, they propose Weight Dilation (WD) that maximally dilates the unsaturated in-channel weights to a constrained range through a mathematically equivalent scaling.  Then they propose  Temporal Parallel Quantizer (TPQ), sets time-step quantization parameters and supports parallel quantization for different time steps. Finally, they propose Block-wise Knowledge Distillation (BKD) to align the quantized models with the full-precision models at a block level.

**Strengths:**

The authors have conducted extensive experiments on diffusion model tasks. We can see the proposed methods performs consistently well as compared to SOTA baselines. Besides, they conducted ablation studies to check the impact of each component and do efficiency analysis.

**Weaknesses:**

1) Ideas of scaling from activations to weights have been proposed in other papers.

2) Abalation studies is not convincing, details can be seen in Questions section.

3) I am doubting whether the module of BKD can be applied to large-scale diffusion models without training.

I will improve my score if the reviewers can address my concern.

**Questions:**

1) For the scaling from activation to weight idea, it has been proposed in the paper AWQ: Activation-aware Weight Quantization for LLM Compression and Acceleration. It would be good if can compare with this paper.

2) For the ablation studies in table 3, it lacks of one combination: WD(Yes) TPQ(No) BKD(Yes). Besides, comparing last two rows, seems the module WD doesn't bring significant contributions. This makes me doubt about the contributions of both WD and TPQ to the final performance. I just doubt maybe the BKD contributes most and nearly all to the performance improvement. Can you conduct one more experiment on the combination:  WD(No) TPQ(No) BKD(Yes)? Thanks.

3) For table 4, can you compare with baselines mentioned in Table 2 for Efficiency comparisons? The baselines in Table 4 seems out-dated.

---

> ### Author Response · Authors · 2024-11-22
>
> We thank the reviewer for the positive review and constructive comments. We provide our responses as follows.
> >**Q1: Ideas of scaling from activations to weights have been proposed in other papers. It would be good if can compare with AWQ.**
>
> **A1**: By considering your valuable comment, we replied your concern in overall rebuttal (1). Our scaling method is tailored to the unique activation-weight distribution of DMs, making it substantially different from previous approaches. In "Appendix G'' of the paper, we conduct detailed experiments comparing our scaling method with previous scaling approaches, including **AWQ**. The results show that applying existing scaling methods actually degrades performance, whereas our method further enhances it.
> >**Q2: Ablation study is not convincing, which lacks of one combination: WD(Yes) TPQ(No) BKD(Yes). The reviewer doubts maybe the BKD contributes most and nearly all to the performance improvement.**
>
> **A2**: Thank you for your valuable suggestion. We replied to your concern in the overall rebuttal (3). Due to retraining the quantized weights, BKD transforms our method into a QAT framework to achieve more significant performance improvements. The proposed TPQ and WD further improve the FID score↓ from **18.26** to **9.13**, achieving a usable generation effect. Without BKD, TPQ and WD improve the FID score of PTQ methods from **120.24** to **16.27**. Compared to the latest PTQ work (TFMQ-DM [1]), which achieved an FID score of **258.81**, TPQ and WD significantly push the performance limits of PTQ methods.
> >**Q3: The reviewer doubts whether the module of BKD can be applied to large-scale diffusion models without training?**
>
> **A3**:Thank you for your constructive question. Currently, applying BKD to large diffusion models still requires training.
> However, due to its QAT-based approach with shorter backpropagation paths, it offers notable advantages in both efficiency and accuracy compared to other training methods.
> In the future, we will further investigate the application of BKD to large-scale diffusion models.
> >**Q4: The baselines in Table 4 of the paper seems out-dated. Can you compare with baselines mentioned in Table 2 of the paper for Efficiency comparisons?**
>
> **A4**: Following your suggestion, we compare our method with the latest QAT-based method (QuEST [2]) from the "Table 2" of the paper. As shown in Table, our method outperforms QuEST in terms of efficiency (calibration samples, time cost, and GPU memory). Thanks to your valuable suggestion again, we will include this comparison in the updated version.
>
> | Method | Frameworks | Calib. | Training Data | Time Cost | GPU Memory | FID↓ |
> |:------------:|:------------:|:------------:|:---------------:|:---------------:|:---------------:|:------------:|
> | QuEST | V-QAT | 5120 | 0 | 15.25 h | 20642 MB | 38.43 |
> | Ours | V-QAT | 1024 | 0 | 6.56 h | 14680 MB | 8.01 |
>
> Reference:
> [1] TFMQ-DM: Temporal Feature Maintenance Quantization for Diffusion Models. CVPR 2024.
> [2] Quest: Low-bit diffusion model quantization via efficient selective finetuning.
>
> ---
> Thank you again for helping us improve the paper and hope our response can resolve your concerns. Please let us know if you have any further questions. We will be actively available until the end of rebuttal period. If you feel your concerns are addressed, please consider reevaluating our work. Looking forward to hearing from you.

---

> ### Author Response · Authors · 2024-11-25
> **Kindly Inquiry to Reviewer sQQb**
>
> Dear Reviewer sQQb,
>
> We greatly appreciate the time and effort in reviewing our work. We have carefully considered your comments and suggestions and have made significant revisions to address the concerns you raised. We are eager to ensure that our paper meets the high standards of our respected reviewers.
>
> We understand your busy schedule, but given that the author-reviewer discussion stage is coming to an end, we sincerely hope that your concerns can be addressed promptly. Please don’t hesitate to let us know if there is any additional feedback you might have at this stage.
>
> Best regards,
> Authors of #2486.

---

> ### Author Response · Authors · 2024-12-01
> **Further Inquiry to Reviewer sQQb**
>
> Dear Reviewer sQQb,
>
> We sincerely apologize for disturbing you again. We are truly grateful for the time and effort you have dedicated to reviewing our work. Your constructive comments and feedback have made a significant contribution to improving our work.
> Based on your suggestions, we have made the following improvements in our new submission:
> - We have provided clearer descriptions to highlight the differences between our approach and previous scaling methods. And we demonstrate the advantages and novelty of our method through both theoretical analysis and experimental results, compared to previous methods (including AWQ).
> - We have included comprehensive ablation studies that clearly showcase the effectiveness of each component of our approach.
> - Following your suggestion, we have expanded our efficiency evaluation to include comparisons with the latest methods, further illustrating the superior efficiency of our approach.
>
> With the sincere intention of addressing all the reviewer’s concerns, we would like to kindly confirm whether there are any remaining issues with our work. We will promptly respond to any feedback you may have before the author-reviewer discussion phase concludes, so please feel free to express any questions. If you feel your concerns are addressed, please consider reevaluating our work. Once again, thank you for your thoughtful review, and we look forward to your reply.
>
> Best regards,
> Authors of #2486.

---

> > ### Comment · Reviewer_sQQb · 2024-12-02
> > **Most concerns are addressed**
> >
> > As most of my concerns are addressed, I have up my score. Thanks for the efforts.

---

> > > ### Author Response · Authors · 2024-12-02
> > > **Thanks to your feedback**
> > >
> > > Dear Reviewer sQQb,
> > >
> > > Thank you for your time and feedback. We truly appreciate your careful consideration of our responses and valuable contributions to our work.
> > >
> > > Best regards,
> > > Authors of #2486

---

### Author Response · Authors · 2024-11-22
**Overall Rebuttal by Author**

Dear Reviewers and Area Chairs,

We would like to thank all the reviewers for their constructive feedback and patient review. Here, we respond to three common questions about our work:

(1). Ideas of scaling from activations to weights have been proposed in other papers.

(2). The contributions of TPQ and BKD appear to be modest.

(3). Ablation studies are inadequate.

As the response includes figures, we provide an anonymous, non-expiring [PDF link](https://anonymous.4open.science/r/ICLR-rebuttal/overall-rebuttal.pdf). We kindly request the Reviewers and the Area Chairs to refer to the PDF for details.

---

### Comment · Area_Chair_v3dp · 2024-12-02

Dear reviewers,

Could you please take a look at the author responses and let the authors know if your concerns have been addressed or not? Thanks very much!

Dear Reviewer umqh,

The authors provided additional responses to your unresolved concerns. You may also take a look. Thank you very much!

Best regards,

AC

---

### Meta-Review · Area_Chair_v3dp · 2024-12-21

**Metareview:**

The paper presents DilateQuant, a quantization framework for diffusion models featuring three key components: Weight Dilation (WD), Temporal Parallel Quantizer (TPQ), and Block-wise Knowledge Distillation (BKD). The overall objective is to improve quantization efficiency while maintaining high accuracy. The paper is well-structured and easy to follow. However, there are significant concerns about the novelty, technical contributions, and experimental validations. For example, the WD approach is closely aligned with existing methods like SmoothQuant and channel-wise scaling techniques. TPQ and BKD are also incremental contributions that follow established techniques in time-step-specific quantization and block-wise distillation. For these limitations, I would like to recommend rejecting this paper.

**Additional Comments On Reviewer Discussion:**

During the rebuttal period, 2 out of 4 reviewers responded to the authors’ replies. Reviewer sQQb was satisfied with the rebuttal and increased the score, while Reviewer umqh still had the concerns that the differences between TPQ and prior methods are minimal.

---

### Decision · Program_Chairs · 2025-01-22

Reject